

# Simple analytical–statistical models (ASMs) for mean annual permafrost table temperature and active-layer thickness estimates

Tomáš Uxa[1,2], Filip Hrbáček[2], and Michaela Kňažková[2]

[1]Institute of Geophysics, Czech Academy of Sciences, Prague, Czech Republic
[2]Polar-Geo-Lab, Department of Geography, Faculty of Science, Masaryk University, Brno, Czech Republic

**Correspondence:** Tomáš Uxa (uxa@ig.cas.cz)

**Abstract.** A variety of numerical, analytical and statistical models have been developed for estimating the mean annual permafrost table temperature (MAPT) and active-layer thickness (ALT). These tools typically require at least a few ground physical properties, such as thermal conductivity, heat capacity, water content or bulk density, as input parameters in addition to temperature variables, which are, however, unavailable or unrepresentative at most sites. Ground physical properties are therefore commonly estimated, which may yield model outputs of unknown validity. Hence, we devised two simple analytical–statistical models (ASMs) for estimating MAPT and ALT, which are driven solely by pairwise combinations of thawing and freezing indices in the active layer; no ground physical properties are required. ASMs reproduced MAPT and ALT well in most numerical validations, which corroborated their theoretical assumptions under idealized scenarios. Under field conditions of Antarctica and Alaska, the mean ASMs deviations in MAPT and ALT were less than 0.03 °C and 5 %, respectively, which is similar or better than other analytical or statistical models. This suggests that ASMs can be useful tools for estimating MAPT and ALT under a wide range of climates and ground physical conditions.

## 1 Introduction

Of ∼11 % of the Earth's exposed land surface underlain by permafrost (Obu, 2021), most seasonally thaws from the ground surface to a depth of up to several meters and then completely refreezes (active layer), which is mainly controlled by climate conditions and ground physical properties (Bonnaventure and Lamoureux, 2013). The active layer greatly influences the energy and mass transfer between the underlying permafrost, ground surface and the atmosphere, and is therefore critical for the dynamics of hydrologic, geomorphic, pedogenic, biologic and biogeochemical processes including greenhouse gas fluxes, as well as for human infrastructure in permafrost regions (e.g., Grosse et al., 2016; Walvoord and Kurylyk, 2016; Hjort et al., 2022). As climate is a first-order control on ground temperatures and thaw depth (Wang et al., 2019; Smith et al., 2022), the thermal state of permafrost and the thickness of the active layer have attracted a huge interest over recent decades because they are important measures of how the climate system is evolving (Li et al., 2022; Hrbáček et al., 2023b). Besides that, climate changes have provoked permafrost warming and active-layer thickening at a global scale (Biskaborn et al., 2019; Noetzli et al., 2024), which can have severe consequences on landscape and ecosystem stability as well as infrastructure integrity. Carbon release due to permafrost degradation is likely to trigger feedback mechanisms with impacts on the Earth's climate system



(Lawrence et al., 2015; Schuur et al., 2022). The permafrost and active-layer monitoring is therefore of utmost scientific and societal importance (Brown et al., 2000; Biskaborn et al., 2015).

The thermal state of permafrost and the thickness of the active layer have commonly been investigated by semi-continuous temperature measurements using data loggers with temperature sensors distributed in vertical arrays across the active layer and near-surface permafrost (e.g., Biskaborn et al., 2015; Noetzli et al., 2021), by periodic or semi-continuous geophysical
measurements using electric, electromagnetic or seismic methods (e.g., Hauck, 2002; Farzamian et al., 2020), or by periodic thaw-depth measurements using physical probing with rigid rods or thaw-tube readings (e.g., Burn, 1998; Bonnaventure and Lamoureux, 2013). Of these methods, temperature measurements using data loggers are the most convenient in terms of accuracy, temporal resolution and/or logistics, which is well suitable for frequently remote and poorly accessible permafrost regions that have limited or no technical infrastructure (Brown et al., 2000; Biskaborn et al., 2015). At many places, however, temper-
atures are only measured in the active layer, and the permafrost temperatures and the active-layer thickness must therefore be estimated in these situations. This has been done using either statistical methods or numerical and analytical models of various complexity (e.g., Riseborough et al., 2008; Bonnaventure and Lamoureux, 2013; Aalto et al., 2018).

Of these solutions, analytical models in particular have become widely popular for estimating the mean annual temperature at the base of the active layer or the top of permafrost (hereafter referred to as the mean annual permafrost table temperature,
MAPT) (Garagulya, 1990; Romanovsky and Osterkamp, 1995; Smith and Riseborough, 1996) and the active-layer thickness (ALT) (Neumann, c. 1860; Stefan, 1891; Kudryavtsev et al., 1977) because of their simplicity, small number of input parameters, computational efficiency and yet sufficient accuracy, which is highly advantageous for diverse permafrost regions and environmental settings (e.g., Anisimov et al., 1997; Nelson et al., 1997; Zhao et al., 2017; Obu et al., 2019, 2020). However, these tools require at least a few ground physical properties, such as thermal conductivity, heat capacity, water content or bulk
density, as input parameters in addition to temperature variables, which are seldom available at most sites. Ground physical properties are therefore commonly estimated, which may yield model outputs of unknown validity. But even in-situ measurements of ground physical properties may not guarantee accurate model outputs either, as they are usually taken annually or less frequently and are then typically treated as constants in models, regardless of their temporal variability (e.g., Gao et al., 2020; Hrbáček et al., 2023a; Li et al., 2023; Kňažková and Hrbáček, 2024; Wenhao et al., 2024).
Here, we devise two novel analytical–statistical models (ASMs) for MAPT and ALT, which are driven solely by thawing and freezing indices at two distinct depths in the active layer to address the general lack and/or non-representativeness of ground physical data for permafrost models. We test these solutions against numerical model simulations for idealized scenarios as well as against field observations from distinct permafrost environments of Antarctica and Alaska, and we discuss their performance, advantages and limitations.



## 2 Model derivations

### 2.1 Mean annual permafrost table temperature

Besides other solution (Garagulya, 1990), MAPT [°C] can be calculated by the TTOP model (Romanovsky and Osterkamp, 1995; Smith and Riseborough, 1996), which assumes that the ratio of thawed and frozen thermal conductivity and the effects of latent heat produce the difference between MAPT and the mean annual ground surface temperature (thermal offset). The TTOP formula for permafrost conditions (MAPT $\leq 0\,°\text{C}$) is as follows (Romanovsky and Osterkamp, 1995; Smith and Riseborough, 1996)

$$\text{MAPT} = \frac{\frac{k_t}{k_f} I_{ts} - I_{fs}}{P}, \tag{1}$$

where $k_t$ [W m$^{-1}$ K$^{-1}$] and $k_f$ [W m$^{-1}$ K$^{-1}$] is the thawed and frozen thermal conductivity, respectively, $I_{ts}$ [°C d] and $I_{fs}$ [°C d] is the ground surface thawing and freezing index, respectively (both expressed degree-days and in absolute values), and $P$ [365 d] is the length of one year.

Besides surface temperatures, Eq. (1) is valid for temperatures measured at any depth in the active layer, which is highly convenient because ground surface temperature is difficult to measure due to surface radiative and convective energy fluxes and due to problematic fixing of temperature sensors exactly at the ground surface level (Riseborough, 2003). Hence, MAPT based on temperatures measured at two distinct depths in the active layer $z_1$ and $z_2$ ($z_1 < z_2 < \text{ALT}$) can be expressed as follows

$$\text{MAPT} = \frac{\frac{k_t}{k_f} I_{tz_1} - I_{fz_1}}{P}, \tag{2}$$

$$\text{MAPT} = \frac{\frac{k_t}{k_f} I_{tz_2} - I_{fz_2}}{P}, \tag{3}$$

where $I_{tz_1}$ [°C d] and $I_{fz_1}$ [°C d] is the thawing and freezing index, respectively, at the depth $z_1$, and $I_{tz_2}$ [°C d] and $I_{fz_2}$ [°C d] is the thawing and freezing index, respectively, at the depth $z_2$. This implies that Eq. (2) and (3) are equivalent:

$$\frac{\frac{k_t}{k_f} I_{tz_1} - I_{fz_1}}{P} = \frac{\frac{k_t}{k_f} I_{tz_2} - I_{fz_2}}{P}. \tag{4}$$

Solving Eq. (4) for the thermal conductivity ratio yields

$$\frac{k_t}{k_f} = \frac{I_{fz_1} - I_{fz_2}}{I_{tz_1} - I_{tz_2}}. \tag{5}$$

Equation (5) can be then substituted for the thermal conductivity ratio in Eq. (2) and (3) as follows

$$\text{MAPT} = \frac{\frac{I_{fz_1} - I_{fz_2}}{I_{tz_1} - I_{tz_2}} I_{tz_1} - I_{fz_1}}{P}, \tag{6}$$

$$\text{MAPT} = \frac{\frac{I_{fz_1} - I_{fz_2}}{I_{tz_1} - I_{tz_2}} I_{tz_2} - I_{fz_2}}{P}. \tag{7}$$



Subsequently, Eq. (6) and (7) both simplify to the same formula for MAPT:

$$\text{MAPT} = \frac{\frac{I_{fz_1} I_{tz_2} - I_{fz_2} I_{tz_1}}{I_{tz_1} - I_{tz_2}}}{P}. \tag{8}$$

Substantially, Eq. (8) implies that MAPT can be simply estimated using thawing and freezing indices at two distinct depths in the active layer alone, that is, without the knowledge of the thermal conductivity ratio.

While Eq. (8) has a physical basis, it can be shown that it is in principle a linear extrapolation of the freezing index to the depth where the thawing index becomes zero, with the slope defined by the thermal conductivity ratio, and its division by the length of one year. Using the same notation as before, this can be expressed as follows

$$\frac{I_{fz_1} - I_{f_{\text{ALT}}}}{I_{tz_1} - I_{t_{\text{ALT}}}} = \frac{I_{fz_1} - I_{fz_2}}{I_{tz_1} - I_{tz_2}}, \tag{9}$$

$$\frac{I_{fz_2} - I_{f_{\text{ALT}}}}{I_{tz_2} - I_{t_{\text{ALT}}}} = \frac{I_{fz_1} - I_{fz_2}}{I_{tz_1} - I_{tz_2}}, \tag{10}$$

where $I_{t_{\text{ALT}}}$ [°C d] and $I_{f_{\text{ALT}}}$ [°C d] represents the thawing and freezing index at the base of the active layer. Solving Eq. (9) and (10) for $I_{f_{\text{ALT}}}$ gives

$$-I_{f_{\text{ALT}}} = \frac{I_{fz_1} - I_{fz_2}}{I_{tz_1} - I_{tz_2}} \left( I_{tz_1} - I_{t_{\text{ALT}}} \right) - I_{fz_1}, \tag{11}$$

$$-I_{f_{\text{ALT}}} = \frac{I_{fz_1} - I_{fz_2}}{I_{tz_1} - I_{tz_2}} \left( I_{tz_2} - I_{t_{\text{ALT}}} \right) - I_{fz_2}. \tag{12}$$

Since the thawing index at the base of the active layer is zero, Eq. (11) and (12) become equivalent to Eq. (6) and (7), respectively, when divided by the length of one year, and both simplify to Eq. (8). This documents that Eq. (8) for MAPT is analytical and statistical at the same time because it integrates both approaches.

## 2.2 Active-layer thickness

Besides other solutions (Neumann, c. 1860; Kudryavtsev et al., 1977), ALT [m] can be calculated by the Stefan (1891) model, which builds on the premise that the conductive heat flux above the thaw front equals to the rate at which latent heat is absorbed as the thaw front propagates downwards. Its simplest is as follows (Lunardini, 1981)

$$\text{ALT} = \sqrt{\frac{2 k_t I_{ts}}{L \phi}}, \tag{13}$$

where $L$ [$3.34 \times 10^8$ J m$^{-3}$] is the volumetric latent heat of fusion of water and $\phi$ [–] is the volumetric water content. Note that the thawing index must be multiplied by the scaling factor of $86\,400$ s d$^{-1}$ in the Stefan model to yield correct outputs. As stated previously (Sect. 2.1), the ground surface temperature is difficult to measure (Riseborough, 2003), and therefore the Stefan model has commonly been forced by temperatures recorded at some depth in the active layer. However, this has rarely been accounted for, although it has been shown to substantially affect the model outputs (Hrbáček and Uxa, 2020; Kaplan Pastíriková et al., 2023), and can be easily implemented as follows (Riseborough, 2003; Hayashi et al., 2007)

$$\text{ALT} = z + \sqrt{\frac{2 k_t I_{tz}}{L \phi}}, \tag{14}$$



where $z$ [m] represents the depth where the forcing temperature was measured and $I_{tz}$ [°C d] is the thawing index at the depth $z$. ALT estimated using thawing indices measured at two distinct depths in the active layer $z_1$ and $z_2$ ($z_1 < z_2 <$ ALT) can be expressed as follows

$$\text{ALT} = z_1 + \sqrt{\frac{2k_t I_{tz_1}}{L\phi}}, \tag{15}$$

$$\text{ALT} = z_2 + \sqrt{\frac{2k_t I_{tz_2}}{L\phi}}. \tag{16}$$

This implies that Eq. (15) and (16) are equivalent:

$$z_1 + \sqrt{\frac{2k_t I_{tz_1}}{L\phi}} = z_2 + \sqrt{\frac{2k_t I_{tz_2}}{L\phi}}. \tag{17}$$

The vertical distance between $z_2$ and $z_1$ can be expressed as

$$z_2 - z_1 = \sqrt{\frac{2k_t I_{tz_1}}{L\phi}} - \sqrt{\frac{2k_t I_{tz_2}}{L\phi}}, \tag{18}$$

which simplifies to

$$z_2 - z_1 = \sqrt{\frac{2k_t}{L\phi}} \left( \sqrt{I_{tz_1}} - \sqrt{I_{tz_2}} \right). \tag{19}$$

Subsequently rearranging Eq. (19) gives

$$\frac{z_2 - z_1}{\sqrt{I_{tz_1}} - \sqrt{I_{tz_2}}} = \sqrt{\frac{2k_t}{L\phi}}, \tag{20}$$

where the right-hand side corresponds to the so-called edaphic term, which has previously been used in numerous studies (Nelson and Outcalt, 1987; Hinkel and Nicholas, 1995; Nelson et al., 1997; Anisimov et al., 2002; Shiklomanov and Nelson, 2002; Shiklomanov et al., 2010; de Pablo et al., 2018; Peng et al., 2023) to combine the ground physical properties in the Stefan model into a single variable as follows

$$\text{ALT} = E\sqrt{I_{tz}}, \tag{21}$$

where $E$ [m s$^{-0.5}$ K$^{-0.5}$] denotes the edaphic term given by

$$E = \sqrt{\frac{2k_t}{L\phi}}. \tag{22}$$

Usually, Eq. (21) has been referred to as the modified Stefan model and proved to be useful in situations where the ground physical properties were unavailable and/or for spatial modelling of ALT (Nelson and Outcalt, 1987; Hinkel and Nicholas, 1995; Nelson et al., 1997; Anisimov et al., 2002; Shiklomanov and Nelson, 2002; Shiklomanov et al., 2010; Peng et al., 2023).





Its major advantage is that it can largely overcome many of the shortcomings of the simplistic Stefan model (Eq. 13), which assumes that the ground physical properties throughout the active layer are constant, the active-layer temperature decreases linearly from the surface to the bottom frozen layer that is at $0\,^{\circ}\mathrm{C}$, and the conductive heat flux is fully consumed by latent heat to thaw the active layer (Kurylyk, 2015). However, the value of the edaphic term has only been derived based on empirical
relationships between ALT and thawing index in several thawing seasons and/or at multiple locations (Nelson et al., 1997; Anisimov et al., 2002; Shiklomanov and Nelson, 2002; Peng et al., 2023). This led on the one hand to its high accuracy for the calibration conditions, but on the other hand had limitations in terms of its transferability to other thawing seasons and/or locations. Notwithstanding that, the edaphic term can be implemented in Eq. (15) and (16) as follows

$$\mathrm{ALT} = z_1 + E\sqrt{I_{tz_1}}, \tag{23}$$

$$\mathrm{ALT} = z_2 + E\sqrt{I_{tz_2}}. \tag{24}$$

Substituting the left-hand side of Eq. (20) for the edaphic term in Eq. (23) and (24) yields

$$\mathrm{ALT} = z_1 + \frac{z_2 - z_1}{\sqrt{I_{tz_1}} - \sqrt{I_{tz_2}}}\sqrt{I_{tz_1}}, \tag{25}$$

$$\mathrm{ALT} = z_2 + \frac{z_2 - z_1}{\sqrt{I_{tz_1}} - \sqrt{I_{tz_2}}}\sqrt{I_{tz_2}}. \tag{26}$$

Simplifying Eq. (25) and (26) then produces the same formula for ALT:

$$\mathrm{ALT} = \frac{z_2\sqrt{I_{tz_1}} - z_1\sqrt{I_{tz_2}}}{\sqrt{I_{tz_1}} - \sqrt{I_{tz_2}}}. \tag{27}$$

Substantially, Eq. (27) implies that ALT can be simply estimated using thawing indices at two distinct depths in the active layer alone, that is, without the knowledge of the ground physical properties or the edaphic term.

While Eq. (27) has a physical basis, it can also be shown that it is in principle a linear extrapolation of the depth at which the square root of the thawing indices becomes zero (cf. Riseborough, 2003), with the slope defined by the edaphic term. Using
the same notation as before, this can be expressed as follows

$$\frac{\mathrm{ALT} - z_1}{\sqrt{I_{tz_1}} - \sqrt{I_{t_{\mathrm{ALT}}}}} = \frac{z_2 - z_1}{\sqrt{I_{tz_1}} - \sqrt{I_{tz_2}}}, \tag{28}$$

$$\frac{\mathrm{ALT} - z_2}{\sqrt{I_{tz_2}} - \sqrt{I_{t_{\mathrm{ALT}}}}} = \frac{z_2 - z_1}{\sqrt{I_{tz_1}} - \sqrt{I_{tz_2}}}. \tag{29}$$

Solving Eq. (28) and (29) for ALT gives

$$\mathrm{ALT} = z_1 + \frac{z_2 - z_1}{\sqrt{I_{tz_1}} - \sqrt{I_{tz_2}}}\left(\sqrt{I_{tz_1}} - \sqrt{I_{t_{\mathrm{ALT}}}}\right), \tag{30}$$

$$\mathrm{ALT} = z_2 + \frac{z_2 - z_1}{\sqrt{I_{tz_1}} - \sqrt{I_{tz_2}}}\left(\sqrt{I_{tz_2}} - \sqrt{I_{t_{\mathrm{ALT}}}}\right). \tag{31}$$

Since the thawing index at the base of the active layer is zero, Eq. (30) and (31) are equivalent to Eq. (25) and (26), respectively, and both simplify to Eq. (27). As with Eq. (8), this documents that Eq. (27) for ALT is analytical and statistical at the same time because it integrates both approaches.





## 3 Model validations

The validity of ASMs for estimating MAPT and ALT given by Eq. (8) and (27), respectively, was tested in a twofold manner, with ground temperatures simulated by a simple one-dimensional numerical model for idealized scenarios and those from field observations.

### 3.1 Idealized scenarios

We considered five scenarios with a mean annual air temperature (MAAT) of $-12\,°\text{C}$, $-10\,°\text{C}$, $-8\,°\text{C}$, $-6\,°\text{C}$ and $-4\,°\text{C}$ that

varied sinusoidally over a year within a range of $40\,°\text{C}$. The air temperatures were converted to ground surface temperature series using linear scaling with so-called thawing and freezing $n$-factors of 1 and 0.5, respectively (Lunardini, 1978). Ground temperatures were then simulated using a one-dimensional numerical model by solving the transient heat conduction equation with phase changes (Carslaw and Jaeger, 1959):

$$C_{\text{eff}}\frac{\partial T}{\partial t} = \frac{\partial}{\partial z}\left(k\frac{\partial T}{\partial z}\right), \tag{32}$$

where $C_{\text{eff}}$ [J m$^{-3}$ K$^{-1}$] is the apparent volumetric heat capacity, $T$ [°C] is the temperature, $t$ [s] is the time, and $k$ [W m$^{-1}$ K$^{-1}$] is the thermal conductivity. Ground was set to be fully frozen and thawed at $T_f$ [$-0.05\,°\text{C}$] and $T_t$ [$0.05\,°\text{C}$], respectively, and linear intermediate in between. Although simplistic, this was chosen to be as close as possible to ASMs, which assume a water—ice transition at $0\,°\text{C}$, while ensuring numerical stability. Similar to Sun et al. (2020), the apparent volumetric heat capacity and thermal conductivity accounted for phase changes with latent heat effects as follows

$$C_{\text{eff}} = \begin{cases} C_f & \text{for } T \leq T_f \\ C_f + (C_t - C_f)\frac{T-T_f}{T_t-T_f} + \frac{L\phi}{T_t-T_f} & \text{for } T_f < T \leq T_t \\ C_t & \text{for } T > T_t \end{cases}, \tag{33}$$

$$k = \begin{cases} k_f & \text{for } T \leq T_f \\ k_f + (k_t - k_f)\frac{T-T_f}{T_t-T_f} & \text{for } T_f < T \leq T_t \\ k_t & \text{for } T > T_t \end{cases}, \tag{34}$$

where $C_f$ [J m$^{-3}$ K$^{-1}$] and $C_t$ [J m$^{-3}$ K$^{-1}$] is the frozen and thawed volumetric heat capacity, respectively. The values of the frozen thermal conductivity and the frozen volumetric heat capacity were estimated from the thawed ones based on the volumetric water content as follows (Nicolsky et al., 2009)

$$k_f = k_t\left(\frac{k_i}{k_w}\right)^{\phi}, \tag{35}$$

$$C_f = C_t - \phi(C_w - C_i), \tag{36}$$

where $k_i$ [2.22 W m$^{-1}$ K$^{-1}$] is the thermal conductivity of ice, $k_w$ [0.57 W m$^{-1}$ K$^{-1}$] is the thermal conductivity of water, $C_w$ [4.21×10$^6$ J m$^{-3}$ K$^{-1}$] is the volumetric heat capacity of water, and $C_i$ [2.05×10$^6$ J m$^{-3}$ K$^{-1}$] is the volumetric heat capacity of ice.





**Table 1.** Values of ground physical properties used in the numerical model simulations for idealized scenarios.

| Variable | Value | Unit |
| --- | --- | --- |
| **Peat** | | |
| Depth | 0–0.2 | m |
| Thawed thermal conductivity | 0.50 | $\mathrm{W\,m^{-1}\,K^{-1}}$ |
| Frozen thermal conductivity | 0.92 | $\mathrm{W\,m^{-1}\,K^{-1}}$ |
| Thawed volumetric heat capacity | $2.300\times10^6$ | $\mathrm{J\,m^{-3}\,K^{-1}}$ |
| Frozen volumetric heat capacity | $1.328\times10^6$ | $\mathrm{J\,m^{-3}\,K^{-1}}$ |
| Volumetric water content | 45 | % |
| **Mineral soil** | | |
| Depth | >0.2 | m |
| Thawed thermal conductivity | 1.50 | $\mathrm{W\,m^{-1}\,K^{-1}}$ |
| Frozen thermal conductivity | 2.26 | $\mathrm{W\,m^{-1}\,K^{-1}}$ |
| Thawed volumetric heat capacity | $2.500\times10^6$ | $\mathrm{J\,m^{-3}\,K^{-1}}$ |
| Frozen volumetric heat capacity | $1.852\times10^6$ | $\mathrm{J\,m^{-3}\,K^{-1}}$ |
| Volumetric water content | 30 | % |

One- and two-layer profiles representing mineral soil alone and 20 cm of peat over mineral soil, respectively, that had constant physical properties except for phase changes were considered in these numerical tests (Table 1), as they aimed to demonstrate the viability of ASMs under idealized conditions. Since ASMs assume a homogeneous profile, the two-layer profile was to examine their behaviour when this condition is not met.

The numerical model was solved using an implicit finite-difference scheme for a 100 m deep domain, which was discretized so that the computation nodes were closely spaced in the active layer and shallow permafrost for the most accurate outputs there, while their density decreased towards the deepest node where the temperature remained stable. Specifically, the node spacing was 0.01 m, 0.1 m, 0.5 m, 1 m, 5 m and 10 m in the depth intervals of 0–2 m, 2–5 m, 5–10 m, 10–20 m, 20–50 m and 50–100 m, respectively. At the upper boundary, the model was forced by the ground surface temperatures. A zero heat flux was set at the lower boundary. The initial temperature was established by Eq. (1) using thawing and freezing indices at the ground surface and at the bottom of the top peat layer for the one- and two-layer profiles, respectively, in order to speed up the time to reach the steady-state conditions throughout the model domain. The model was run for 50 years with a time step of 1 hour to ensure that the simulated temperatures are not affected by the initial conditions. Steady-state MAPT, ALT, and thawing and freezing indices simulated for the last year were then used for numerical validations of ASMs given by Eq. (8) and (27).



**Table 2.** List of the Antarctic and Alaskan sites and the number of years/seasons used for the model validations.

| Site | Latitude [°] | Longitude [°] | Altitude [m asl] | Validation period | Years for MAPT | Seasons for ALT |
|---|---|---|---|---|---|---|
| **James Ross Island** | | | | | | |
| Abernethy Flats | −63.88138 | −57.94832 | 41 | 2013–2020 | 6–6 | 7–7 |
| Berry Hill slopes | −63.80267 | −57.83863 | 56 | 2017–2020 | 3–3 | 3–3 |
| CALM | −63.80190 | −57.88460 | 10 | 2014–2023 | 7–7 | 8–8 |
| Johann Gregor Mendel | −63.80152 | −57.88330 | 10 | 2011–2023 | 10–12 | 11–12 |
| Johnson Mesa | −63.82250 | −57.93280 | 340 | 2012–2023 | 8–11 | 9–11 |
| **McMurdo Sound** | | | | | | |
| Bull Pass | −77.51847 | 161.86269 | 141 | 1999–2022 | 15–22 | 14–22 |
| Granite Harbour | −77.00655 | 162.52561 | 6 | 2007–2017 | 4–4 | 5–5 |
| Marble Point | −77.41955 | 163.68247 | 47 | 1999–2022 | 18–22 | 17–21 |
| **North Slope of Alaska** | | | | | | |
| Atqasuk | 70.45242 | −157.41178 | 22 | 1998–2010 | 6–9 | 8–12 |
| Barrow (site 1) | 71.32242 | −156.61089 | 9 | 1997–2017 | 15–16 | 15–17 |
| Betty Pingo: polygon center | 70.28258 | −148.89347 | 12 | 2006–2022 | 0–9 | 0–9 |
| Betty Pingo: polygon rim | 70.28258 | −148.89347 | 12 | 2006–2012 | 4–7 | 4–7 |
| Westdock (high): polygon center | 70.37039 | −148.56867 | 3 | 2002–2020 | 16–17 | 18–19 |
| Westdock (high): polygon rim | 70.37039 | −148.56867 | 3 | 2003–2020 | 16–17 | 18–18 |
| Westdock (high): polygon trough | 70.37039 | −148.56867 | 3 | 2003–2020 | 9–17 | 11–18 |
| Westdock (low): polygon center | 70.37047 | −148.56561 | 2 | 2004–2011 | 4–4 | 8–8 |
| Westdock (low): polygon trough | 70.37047 | −148.56561 | 2 | 2004–2022 | 1–9 | 6–13 |

## 3.2 Field observations

Ground temperatures were collected for 17 sites situated in permafrost environments on James Ross Island and McMurdo
Sound in Antarctica and on the North Slope of Alaska in the Arctic (Table 2) in order to test ASMs under diverse climates
and ground physical conditions. A total of 142–192 and 162–210 years/seasons (Table 2) with quality-checked observations of
MAPT, ALT, and thawing and freezing indices were available for individual validation scenarios of ASMs given Eq. (8) and
(27), respectively, (see Sect. 3.3). The variability in the number of available years/seasons for the validations (Table 2) was
because in some years/seasons the active layer was thinner than the deepest sensors used in Eq. (8) and (27) and/or due to data
gaps.



### 3.3 Model evaluation

For both numerical and field validations of ASMs, the thawing and freezing indices were calculated as annual sums of positive and negative mean daily ground temperatures, respectively, and for convenience expressed in degree-days and in absolute values. textrmALT was derived as the maximum seasonal depth of the $0\,°C$ isotherm by a linear interpolation of the depths

where the mean daily ground temperatures were just above and below $0\,°C$. Subsequently, the mean annual temperatures at the same depths were used to interpolate MAPT. We used three pairwise combinations of thawing and freezing indices at the depth of $5\,cm$, $30\,cm$ and $50\,cm$ as inputs of Eq. (8) and (27) for numerical validations, while thawing and freezing indices from the depth intervals of $0$–$10\,cm$, $25$–$35\,cm$ and $45$–$55\,cm$ (for convenience hereafter also referred to as $5\,cm$, $30\,cm$ and $50\,cm$) were considered for field validations because the sensor depths differ at individual sites. However, this did not compromise

the consistency of field validations and allowed us to reveal which depth combinations and in which portion of the active layer worked best. The ASMs outputs were compared with MAPT and ALT from the numerical model simulations and field observations and evaluated using common error metrics, such as the mean error (ME), the mean percentage error (MPE), the mean absolute error (MAE), the mean absolute percentage error (MAPE), and the root-mean-square error (RMSE).

## 4   Results

### 4.1   Mean annual permafrost table temperature

#### 4.1.1   Numerical validation

The numerical model simulations for the five MAAT scenarios showed that the thawing and freezing indices tend to decrease exponentially from the ground surface towards the base of the active layer where the thawing indices are zero (Fig. 1). However, the relationships between the thawing and freezing indices themselves are linear within each subsurface layer (both peat and

mineral soil), and their slopes are governed by the thermal conductivity ratios in the individual layers (Fig. 2).

MAPT estimated by Eq. (8) based on the numerically modelled thawing and freezing indices at the depth pairs of $5/30\,cm$, $5/50\,cm$ and $30/50\,cm$ for the five MAAT scenarios showed almost perfect agreement with MAPT simulated by the numerical model in the one-layer profiles (Table 3), as ME was $-0.003\,°C$ to $-0.002\,°C$, MAE was $0.002\,°C$ to $0.003\,°C$, and RMSE was $0.002\,°C$ to $0.003\,°C$. The accuracy of Eq. (8) was slightly lower in the two-layer profiles (Table 3), as ME was $-0.105\,°C$ to

$-0.003\,°C$, MAE was $0.003\,°C$ to $0.105\,°C$, and RMSE was $0.004\,°C$ to $0.124\,°C$.

Overall, however, these findings corroborate the theoretical assumptions outlined in Sect. 2.1 and justify ASM given by Eq. (8) for estimating MAPT under the idealized scenarios.

#### 4.1.2   Field validation

MAPT estimated by Eq. (8) based on the thawing and freezing indices at the depth pairs $5/30\,cm$, $5/50\,cm$ and $30/50\,cm$ at

the Antarctic and Alaskan sites yielded the site-weighted ME of $0.02\,°C$ to $0.03\,°C$ compared to the observed MAPT (Fig. 3).





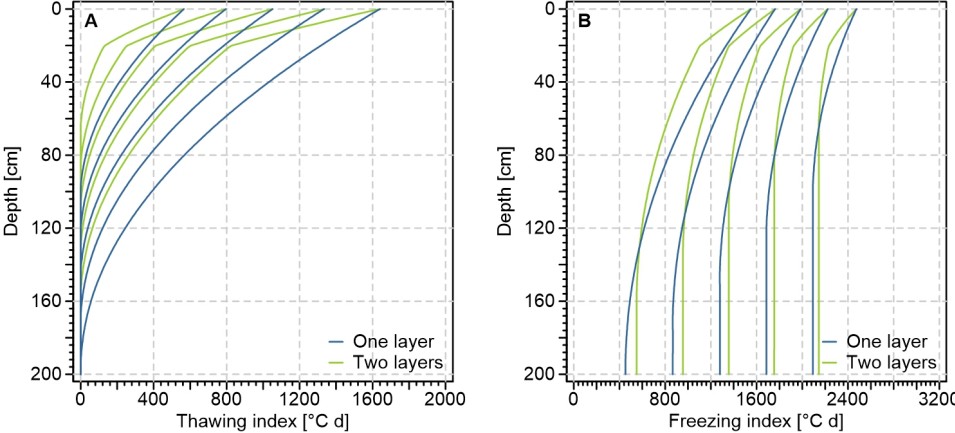

**Figure 1.** Depth profiles of (A) the thawing indices and (B) the freezing indices in the active layer and near-surface permafrost simulated by the numerical model for MAAT of −12 °C, −10 °C, −8 °C, −6 °C and −4 °C that varied sinusoidally over a year within a range of 40 °C. Note the bent shapes of the thawing and freezing indices in the active layer, which only change abruptly at the interface of peat and mineral soil in the two-layer profiles due to distinct physical properties of these materials (see Table 1).

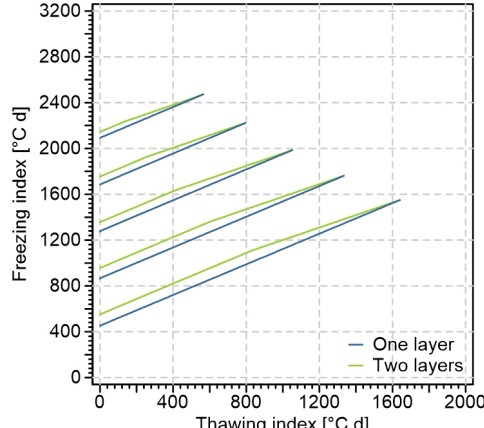

**Figure 2.** Relationships between the thawing and freezing indices in the active layer simulated by the numerical model for MAAT of −12 °C, −10 °C, −8 °C, −6 °C and −4 °C that varied sinusoidally over a year within a range of 40 °C. Note that the relationships are linear, but their slopes change abruptly at the interface of peat and mineral soil in the two-layer profiles due to distinct physical properties of these materials (see Table 1).

Since the errors were scattered around zero (Fig. 3), the site-weighted MAE was somewhat larger of 0.08 °C to 0.14 °C and the site-weighted RMSE was 0.10 °C to 0.17 °C (Fig. 3). The majority of the errors was within ±0.2 °C (Fig. 3).

The accuracy of the ASM estimates was slightly lower in Antarctica (Fig. 3) where the site-weighted ME was −0.04 °C to 0.04 °C, the site-weighted MAE was 0.10 °C to 0.15 °C, and the site-weighted RMSE was 0.13 °C to 0.18 °C. In Alaska, the





**Table 3.** Comparison of MAPT simulated by the numerical model for MAAT of −12 °C, −10 °C, −8 °C, −6 °C and −4 °C that varied sinusoidally over a year within a range of 40 °C and MAPT estimated with ASM given by Eq. (8) based on the numerically modelled thawing and freezing indices at the depth pairs of 5/30 cm, 5/50 cm and 30/50 cm.

| Scenario | MAAT [°C] | $MAPT_{num}$ [°C] | $MAPT_{5/30}$ [°C] | $MAPT_{5/50}$ [°C] | $MAPT_{30/50}$ [°C] |
|---|---|---|---|---|---|
| One layer | −4 | −1.24 | −1.25 | −1.25 | −1.25 |
| | −6 | −2.38 | −2.38 | −2.38 | −2.38 |
| | −8 | −3.50 | −3.51 | −3.51 | −3.51 |
| | −10 | −4.62 | −4.62 | −4.62 | −4.62 |
| | −12 | −5.73 | −5.73 | −5.73 | −5.73 |
| | Mean | −3.49 | −3.50 | −3.50 | −3.50 |
| Two layers | −4 | −1.51 | −1.72 | −1.63 | −1.52 |
| | −6 | −2.62 | −2.77 | −2.70 | −2.62 |
| | −8 | −3.72 | −3.81 | −3.76 | −3.72 |
| | −10 | −4.81 | −4.86 | −4.83 | −4.81 |
| | −12 | −5.88 | −5.90 | −5.88 | −5.88 |
| | Mean | −3.71 | −3.81 | −3.76 | −3.71 |

site-weighted ME was −0.01 °C to 0.09 °C, the site-weighted MAE was 0.07 °C to 0.13 °C, and the site-weighted RMSE was 0.08 °C to 0.15 °C. However, the ASM deviations exhibited very similar distributions in both regions (Fig. 1).

## 4.2   Active-layer thickness

### 4.2.1   Numerical validation

As stated in Sect. 4.1.1, the numerical model simulations for the five MAAT scenarios showed that the thawing indices tend

to decrease exponentially from the ground surface towards the base of the active layer where they are zero (Fig. 1A). If square rooted, however, the bent-shaped depth profiles of the thawing indices become linear within each subsurface layer (both peat and mineral soil), except for subtle deviations near the base of the active layer, and their slopes are governed by the edaphic terms in the individual layers (Fig. 4).

ALT estimated by Eq. (27) based on the numerically modelled thawing indices at the depth pairs of 5/30 cm, 5/50 cm and

30/50 cm for the five MAAT scenarios was well consistent with ALT simulated by the numerical model in the one-layer profiles (Table 4), as ME was 0.8 cm (0.9 %) to 1.8 cm (1.5 %), MAE was 1.6 cm (1.3 %) to 1.8 cm (1.5 %), and RMSE was 1.6 cm to 1.9 cm. On the other hand, the accuracy of Eq. (27) was much worse in the two-layer profiles when the thawing indices from the top peat layer were used for the calculations (Table 4), as ME was −41.0 cm (−35.4 %) to 2.4 cm (2.7 %), MAE was 2.4 cm (2.7 %) to 41.0 cm (35.4 %), and RMSE was 3.4 cm to 35.9 cm. The deviations tended to decrease as the active layer thickened

in the one-layer profiles, while they tended to increase as the active-layer thickened in the two-layer profiles (Table 4).





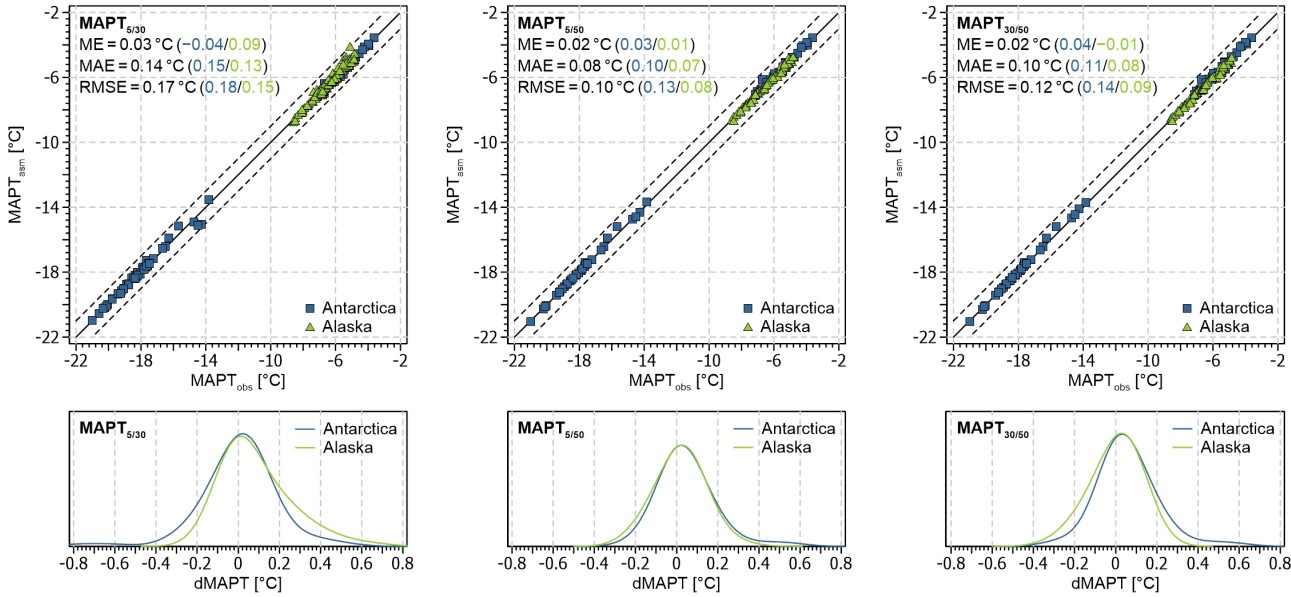

**Figure 3.** (Upper row) Comparison of MAPT observed at the Antarctic and Alaskan sites and MAPT estimated with ASM given by Eq. (8) based on the observed thawing and freezing indices at the depth pairs of 5/30 cm, 5/50 cm and 30/50 cm. The blue and green numbers in parentheses indicate the mean errors for the Antarctic and Alaskan sites, respectively. The black solid and dashed lines represent the line of identity and the deviation of ±1 °C, respectively. (Lower row) Probability distribution of the errors in MAPT estimated with ASM for the depth pairs of 5/30 cm, 5/50 cm and 30/50 cm.

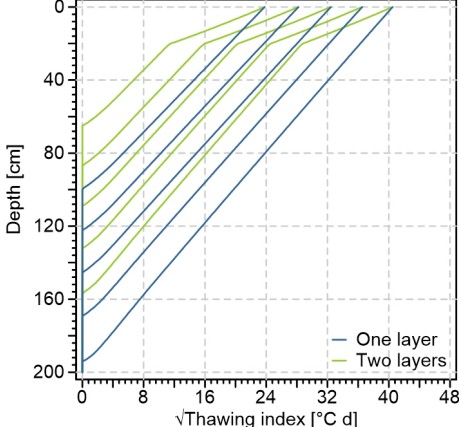

**Figure 4.** Depth profiles of the square-rooted thawing indices in the active layer and near-surface permafrost simulated by the numerical model for MAAT of −12 °C, −10 °C, −8 °C, −6 °C and −4 °C that varied sinusoidally over a year within a range of 40 °C. Note that the bent shapes of the thawing indices (Fig. 1A) become linear when square-rooted, but their slopes change abruptly at the interface of peat and mineral soil in the two-layer profiles due to distinct physical properties of these materials (see Table 1).





**Table 4.** Comparison of ALT simulated by the numerical model for MAAT of $-12\,^\circ$C, $-10\,^\circ$C, $-8\,^\circ$C, $-6\,^\circ$C and $-4\,^\circ$C that varied sinusoidally over a year within a range of $40\,^\circ$C and ALT estimated with ASM given by Eq. (27) based on the numerically modelled thawing and freezing indices at the depth pairs of 5/30 cm, 5/50 cm and 30/50 cm.

| Scenario | MAAT [$^\circ$C] | $ALT_{num}$ [cm] | $ALT_{5/30}$ [cm] | $ALT_{5/50}$ [cm] | $ALT_{30/50}$ [cm] |
|---|---|---|---|---|---|
| One layer | $-4$ | 195 | 193 | 194 | 195 |
| | $-6$ | 170 | 170 | 170 | 171 |
| | $-8$ | 146 | 147 | 148 | 148 |
| | $-10$ | 123 | 125 | 126 | 126 |
| | $-12$ | 100 | 103 | 103 | 103 |
| | Mean | 146.8 | 147.6 | 148.2 | 148.6 |
| Two layers | $-4$ | 157 | 90 | 116 | 158 |
| | $-6$ | 133 | 79 | 102 | 134 |
| | $-8$ | 109 | 69 | 88 | 112 |
| | $-10$ | 87 | 59 | 75 | 90 |
| | $-12$ | 65 | 49 | 62 | 69 |
| | Mean | 110.2 | 69.2 | 88.6 | 112.6 |

Overall, however, these findings corroborate the theoretical assumptions outlined in Sect. 2.2 and justify ASM given by Eq. (27) for estimating ALT under idealized scenarios in one-layer profiles.

### 4.2.2 Field validation

ALT estimated by Eq. (27) based on the thawing indices at the depth pairs of 5/30 cm, 5/50 cm and 30/50 cm at the Antarctic
and Alaskan sites showed the site-weighted ME of $-2.6$ cm ($-4.4$ %) to $-1.4$ cm ($-2.4$ %) compared to the observed ALT (Fig. 5). The site-weighted MAE was somewhat larger, as it attained 4.8 cm (6.9 %) to 8.8 cm (13.5 %), while the site-weighted RMSE was 5.3 cm to 9.8 cm (Fig. 5).

ALT estimates by Eq. (27) were more accurate in Antarctica where the site-weighted ME was 0.9 cm (0.8 %) to 5.4 cm (7.2 %), the site-weighted MAE was 3.5 cm (4.6 %) to 8.4 cm (11.9 %), and the site-weighted RMSE was 4.0 cm to 9.7 cm.
By contrast, in Alaska the site-weighted ME was $-8.6$ cm ($-13.9$ %) to $-3.6$ cm ($-5.6$ %), the site-weighted MAE was 5.2 cm (8.2 %) to 9.1 cm (14.9 %), and the site-weighted RMSE was 5.8 cm to 10.0 cm. The ASM deviations were roughly scattered around zero in Antarctica, while they tended to be negative in Alaska where the deviations also exhibited a bimodal distribution for the depth pair of 5/30 cm (Fig. 5).





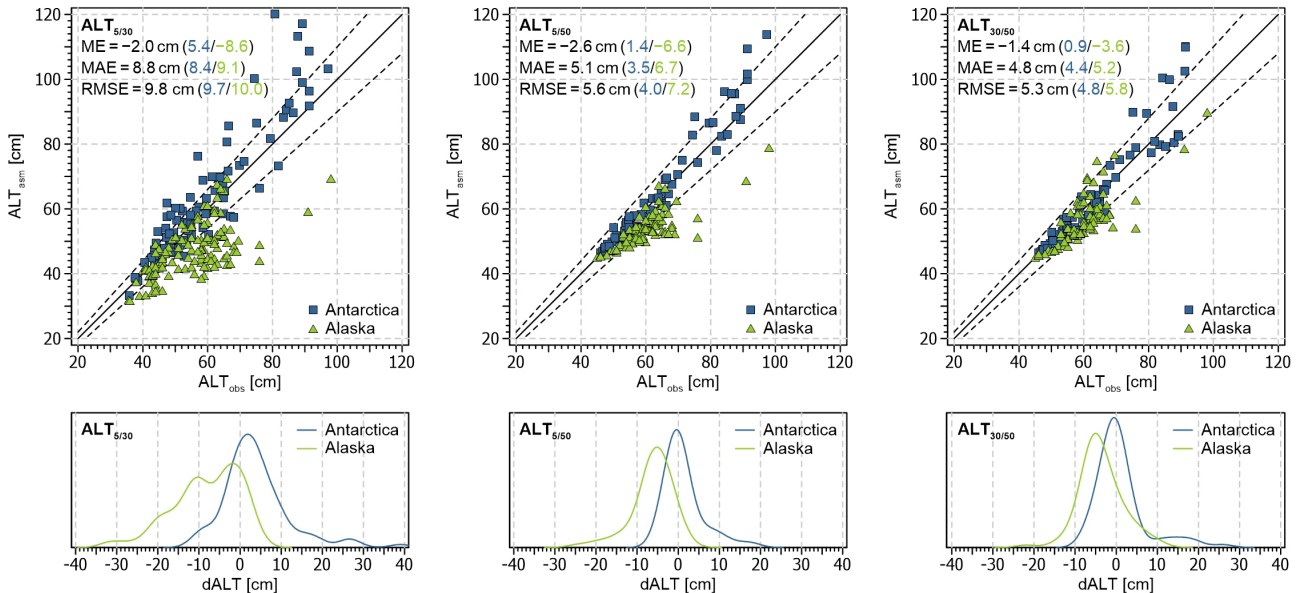

**Figure 5.** (Upper row) Comparison of ALT observed at the Antarctic and Alaskan sites and ALT estimated with ASM given by Eq. (27) based on the observed thawing indices at the depth pairs of 5/30 cm, 5/50 cm and 30/50 cm. The blue and green numbers in parentheses indicate the mean errors for the Antarctic and Alaskan sites, respectively. The black solid and dashed lines represent the line of identity and the deviation of ±10 %, respectively. (Lower row) Probability distribution of the errors in ALT estimated with ASM for the depth pairs of 5/30 cm, 5/50 cm and 30/50 cm.

## 5 Discussion

### 5.1 Model performances

ASMs given by Eq. (8) and (27) reproduced MAPT and ALT with a reasonable accuracy under most idealized scenarios and field conditions, which corroborated their theoretical assumptions (see Sect. 2.1 and 2.2) and suggested that they can work reasonably well under a wide range of climates and ground physical conditions.

### 5.1.1 Mean annual permafrost table temperature

MAPT estimates by Eq. (8) had high accuracy regardless of the stratigraphy of the active layer and the depth pairs used for the calculations (Table 3, Fig. 3). Under idealized scenarios, the ASM deviations in the one-layer profiles were negligible, while in the two-layer profiles the temperatures were underestimated by less than ∼0.1 °C on average (Table 3). Under field conditions, the ASM deviations were close to zero on average, and the majority of them was below ±0.2 °C at the Antarctic and Alaskan sites (Fig. 3), which is within the accuracy of many temperature sensors and similar or better than in most previous studies

that used other analytical or statistical models for MAPT estimates (e.g., Romanovsky and Osterkamp, 1995; Sazonova and Romanovsky, 2003; Ferreira et al., 2017; Way and Lewkowicz, 2018; Wang et al., 2020; Kaplan Pastíriková et al., 2023). This



is likely because the relationship between the thawing and freezing indices is linear within each subsurface layer, and its slope varies rather slightly with vertical changes in ground physical properties at the layer interfaces (Fig. 2). This was noticeable at the Alaskan sites where the presence of peat over mineral soil is common. So far, MAPT models have also typically assumed
that thawed thermal conductivity is lower than frozen one, and that the thermal offset is therefore negativ (e.g., Gisnås et al., 2013; Obu et al., 2019, 2020), which would, however, yield invalid MAPT estimates under reverse conditions. Since Eq. (8) utilizes measured temperatures, it can easily handle even such anomalies, as demonstrated, for example, in McMurdo Sound where the thermal offset is often positive (Lacelle et al., 2016). Additionally, the thermal offset is usually in the order of tenths to first degrees Celsius and decreases exponentially with depth (Goodrich, 1982; Burn and Smith, 1988; Romanovsky and
Osterkamp, 1995). Hence, it was relatively small below the bottom temperature sensors used for the calculations and MAPT estimates were subject to relatively small uncertainties. Somewhat larger deviations in MAPT estimates would, however, be expected in warmer conditions with thicker active layers and high vertical changes in ground physical properties.

### 5.1.2  Active-layer thickness

By contrast, ALT estimates by Eq. (27) had very different accuracy in the one-layer and two-layer profiles that also depended
on the depth pairs used for the calculations (Table 4, Fig. 5). Under idealized scenarios, the ASM deviations in the one-layer profiles were below 1.5 % on average, while in the two-layer profiles the deviations were up to tens of percent, except for the depth pair of 30/50 cm, which excluded the thawing index from the top peat layer with different physical properties (Table 4). The minor deviations in the one-layer profiles and in the two-layer profiles for the depth pair of 30/50 cm were largely because the vertical profiles of the square-rooted thawing indices were not perfectly linear near the base of the active layer (Fig. 4),
which was likely due to upward freezing from the permafrost table at the end of the thawing seasons (cf. Riseborough, 2003). Under field conditions, the ASM deviations were scattered around zero at the Antarctic sites and roughly attained less than 7 % on average, while ALT tended to be underestimated at the Alaskan sites by up to 14 % on average (Fig. 5). Overall, however, the accuracy of ASM given by Eq. (27) was similar or better than in most previous studies that used the other analytical or statistical models for ALT estimates (Anisimov et al., 1997; Nelson et al., 1997; Romanovsky and Osterkamp, 1997; Anisimov
et al., 2002; Shiklomanov and Nelson, 2002; Sazonova and Romanovsky, 2003; Streletskiy et al., 2012; Yin et al., 2016; Zorigt et al., 2016; Hrbáček and Uxa, 2020; Kaplan Pastíriková et al., 2023). The higher accuracy of ASM at the Antarctic sites (Fig. 5) was likely due to the fact that the active layer there is relatively homogeneous in terms of its stratigraphy and physical properties, whereas at the Alaskan sites it typically consists of two distinct layers. This is also why the depth pair of 30/50 cm showed the lowest errors (Fig. 5), as it excluded the surface layer of peat, which is an effective thermal insulator that
substantially alters the temperature gradient in the active layer.

### 5.2  Model advantages

Unlike other analytical or statistical models for estimating MAPT (e.g., Garagulya, 1990; Romanovsky and Osterkamp, 1995; Smith and Riseborough, 1996) and ALT (e.g., Neumann, c. 1860; Stefan, 1891; Kudryavtsev et al., 1977), ASMs given by Eq. (8) and (27) can be utilized in any substrates where conductive heat transfer prevails, such as soil, peat, or solid rock,



without the knowledge of their physical properties. Since ASMs build solely on thawing and freezing indices at two distinct
depths in the active layer, the values of which reflect the rate of heat transfer across their intermediate layer, the solutions
also intrinsically account for the temporal variability of ground physical properties. Likewise, they consider latent and sensible
heat and any other factors that might affect the heat transfer in the active layer, some of which other models do not explicitly
account for. This is highly convenient because data on ground physical properties, such as thermal conductivity, heat capacity,
water content or bulk density, are not readily available at many sites. Ground physical properties for other models estimating
MAPT (e.g., Gisnås et al., 2013; Obu et al., 2019, 2020; Garibaldi et al., 2021) and ALT (e.g., Hinkel and Nicholas, 1995;
Nelson et al., 1997; Anisimov et al., 2002; Shiklomanov and Nelson, 2002) have been set empirically or have been based
on published values, and therefore their values have frequently been of unknown validity. Ground physical properties also
commonly show more or less variability on seasonal and annual time scales (e.g., Gao et al., 2020; Hrbáček et al., 2023a; Li et
al., 2023; Kňažková and Hrbáček, 2024; Wenhao et al., 2024), which most other models cannot handle because they typically
treat ground physical properties as constants.

Another advantage is that ASMs are not limited to temperatures at certain depths, but their inputs can be any depth combi-
nations from within the active layer based on temperature data availability and site characteristics. For best MAPT and ALT
estimates, it is therefore suggested to use thawing and freezing indices from depths as close as possible to the permafrost table,
where available.

Besides field measurements, ASMs can also be forced by diverse climate reanalysis or climate model outputs, if these at
least partially consider the physics of ground thawing and freezing. These products typically provide only ground surface and
shallow active-layer temperatures with limited or no information on ground physical properties, which is frequently insufficient
to determine MAPT and ALT either directly or using conventional solutions. However, this is not an issue for ASMs.

Lastly, ASMs can also be easily reformulated to be used for estimating the mean annual temperature at the base of seasonally
frozen ground and frost depth (see Appendix A and B).

### 5.3   Model limitations

Since ASMs assume homogeneous (one-layer) profiles, they may understate reality in multi-layer profiles that exhibit large
stepwise vertical changes in ground physical properties and/or higher ground-ice contents near the base of the active layer
(Riseborough, 2003). If, for instance, temperature measurements are used only from the top layer, the physical properties of
which differ from those of the layer below, ASMs may therefore be inaccurate (Fig. 2 and  4). Equally, the outputs may have
unknown validity if only shallow temperature measurements in thick active layers are used because they would be based on
the rate of heat transfer in a tiny portion of the active layer, which may differ in its deeper sections (Fig. 2 and  4). On the
other hand, natural variability of ground physical properties with no sharp changes in their vertical distribution is unlikely to
affect ASMs substantially. Other downside of ASMs is that they require temperature measurements at two depths in the active
layer, which may not be available at many sites, and can also be problematic to collect if the active layer is thin. Special care
must also be taken with the depths of the temperature sensors and the vertical distances between them, which must be constant
over time, as well as with the accuracy of the sensors, because any deviations in these may negatively influence the ASMs



outputs. Nevertheless, these issues are largely common to any analytical, statistical and even numerical permafrost models, as
they relate to the quality of the inputs rather than the shortcomings of ASMs themselves.

## 6 Conclusions

We devised two novel ASMs given by Eq. (8) and (27) for estimating MAPT and ALT, respectively, which are driven solely by
pairwise combinations of thawing and freezing indices in the active layer; no ground physical properties are required. ASMs
reproduced MAPT and ALT well under most idealized scenarios, which corroborated their theoretical assumptions. Under
field conditions of Antarctica and Alaska, the mean ASMs deviations in MAPT and ALT were less than $0.03\,°C$ and $5\,\%$,
respectively, which is very promising because it is similar or better than other analytical or statistical models. ASMs worked
best in homogeneous active layers with small vertical changes in ground physical properties and when permafrost table was
close below the temperature sensors considered for MAPT and ALT calculations.

Hence, ASMs for estimating MAPT and ALT can find applications under a wide range of climates and ground physical
conditions wherever at least two temperature measurements in the active layer are available. Besides field measurements,
they can also utilize diverse climate reanalyses or climate model ground temperature products. Lastly, they can be easily
reformulated for estimating the mean annual temperature at the base of seasonally frozen ground and frost depth.

## Appendix A:  Derivation of ASM for mean annual temperature at the base of seasonally frozen ground

Similarly to Eq. (1), the mean annual temperature at the base of seasonally frozen ground (MASFT $> 0\,°C$) is calculated as
follows (Romanovsky and Osterkamp, 1995)

$$\text{MASFT} = \frac{I_{ts} - \frac{k_f}{k_t} I_{fs}}{P},\tag{A1}$$

which has the same attributes as Eq. (1). Hence, MASFT based on temperatures measured at two distinct depths in the season-
ally freezing layer $z_1$ and $z_2$ ($z_1 < z_1 < \text{FD}$) can be expressed as follows

$$\text{MASFT} = \frac{I_{tz_1} - \frac{k_f}{k_t} I_{fz_1}}{P},\tag{A2}$$

$$\text{MASFT} = \frac{I_{tz_2} - \frac{k_f}{k_t} I_{fz_2}}{P}.\tag{A3}$$

This implies that Eq. (A2) and (A2) are equivalent:

$$\frac{I_{tz_1} - \frac{k_f}{k_t} I_{fz_1}}{P} = \frac{I_{tz_2} - \frac{k_f}{k_t} I_{fz_2}}{P}.\tag{A4}$$

Solving Eq. (A4) for the inverse of the thermal conductivity ratio yields

$$\frac{k_f}{k_t} = \frac{I_{tz_1} - I_{tz_2}}{I_{fz_1} - I_{fz_2}}.\tag{A5}$$



Equation (A5) can be then substituted for the thermal conductivity ratio in Eq. (A2) and (A3) as follows

$$\text{MASFT} = \frac{I_{tz_1} - \frac{I_{tz_1} - I_{tz_2}}{I_{fz_1} - I_{fz_2}} I_{fz_1}}{P}, \tag{A6}$$

$$\text{MASFT} = \frac{I_{tz_2} - \frac{I_{tz_1} - I_{tz_2}}{I_{fz_1} - I_{fz_2}} I_{fz_2}}{P}. \tag{A7}$$

Subsequently, Eq. (A6) and (A7) both simplify to the same formula for MASFT:

$$\text{MASFT} = \frac{\frac{I_{fz_1} I_{tz_2} - I_{fz_2} I_{tz_1}}{I_{fz_1} - I_{fz_2}}}{P}, \tag{A8}$$

which only slightly differs from Eq. (A8) and has the same attributes.

## Appendix B: Derivation of ASM for frost depth

Similarly to Eq. (13), frost depth (FD) can be calculated by the Stefan (1891) model as follows

$$\text{FD} = \sqrt{\frac{2k_f I_{fs}}{L\phi}}. \tag{B1}$$

Likewise, note that the freezing index must be multiplied by the scaling factor of $86\,400\ \text{s}\,\text{d}^{-1}$ in the Stefan model to yield correct outputs. FD estimated using freezing indices measured at two distinct depths $z_1$ and $z_2$ ($z_1 < z_1 <$ FD) can be expressed as follows

$$\text{FD} = z_1 + \sqrt{\frac{2k_f I_{fz_1}}{L\phi}}, \tag{B2}$$

$$\text{FD} = z_2 + \sqrt{\frac{2k_f I_{fz_2}}{L\phi}}. \tag{B3}$$

This implies that Eq. (B2) and (B3) are equivalent:

$$z_1 + \sqrt{\frac{2k_f I_{fz_1}}{L\phi}} = z_2 + \sqrt{\frac{2k_f I_{fz_2}}{L\phi}}. \tag{B4}$$

The vertical distance between $z_2$ and $z_1$ can be expressed as

$$z_2 - z_1 = \sqrt{\frac{2k_f I_{fz_1}}{L\phi}} - \sqrt{\frac{2k_f I_{fz_2}}{L\phi}}, \tag{B5}$$

which simplifies to

$$z_2 - z_1 = \sqrt{\frac{2k_f}{L\phi}} \left( \sqrt{I_{fz_1}} - \sqrt{I_{fz_2}} \right). \tag{B6}$$



Subsequently rearranging Eq. (B6) gives

$$\frac{z_2 - z_1}{\sqrt{I_{fz_1}} - \sqrt{I_{fz_2}}} = \sqrt{\frac{2k_f}{L\phi}}, \tag{B7}$$

where the right-hand side corresponds to the edaphic term, which combines the ground physical properties in the Stefan model into a single variable. The edaphic term can be implemented in Eq. (B2) and (B2) as follows

$$FD = z_1 + E\sqrt{I_{fz_1}}, \tag{B8}$$

$$FD = z_2 + E\sqrt{I_{fz_2}}. \tag{B9}$$

Substituting the left-hand side of Eq. (B7) for the edaphic term in Eq. (B8) and (B9) yields

$$FD = z_1 + \frac{z_2 - z_1}{\sqrt{I_{fz_1}} - \sqrt{I_{fz_2}}}\sqrt{I_{fz_1}}, \tag{B10}$$

$$FD = z_2 + \frac{z_2 - z_1}{\sqrt{I_{fz_1}} - \sqrt{I_{fz_2}}}\sqrt{I_{fz_2}}. \tag{B11}$$

Simplifying Eq. (B10) and (B11) then produces the same formula for FD:

$$FD = \frac{z_2\sqrt{I_{fz_1}} - z_1\sqrt{I_{fz_2}}}{\sqrt{I_{fz_1}} - \sqrt{I_{fz_2}}}, \tag{B12}$$

which is the same and has the same attributes as Eq. (27), only the freezing indices are used instead of the thawing ones.

*Data availability.* The validation datasets from James Ross Island are available upon request from Filip Hrbáček (hrbacekfilip@gmail.com), whereas those from McMurdo Sound and North Slope of Alaska can be retrieved from https://www.nrcs.usda.gov/resources/data-and-reports/soil-climate-research-stations.

*Author contributions.* TU: conceptualization, methodology, software, validation, formal analysis, investigation, writing – original draft, visualization. FH: conceptualization, resources, writing – review & editing, supervision, funding acquisition. MK: formal analysis, resources, writing – review & editing.

*Competing interests.* The contact author has declared that none of the authors has any competing interests.

*Acknowledgements.* We acknowledge USDA for access to the soil climate data from the McMurdo Sound and North Slope of Alaska.

*Financial support.* The research was funded by the Czech Science Foundation (project number GM22-28659M).



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
