# Peer review of "Simple analytical–statistical models (ASMs) for mean annual permafrost table temperature and active-layer thickness estimates"

_EGUsphere, 2024_

## Author Comment (AC1)

**AUTHORS' RESPONSE TO THE COMMENTS OF THE REFEREE #1**

**RC1:** GENERAL COMMENTS

The authors propose a simple method for estimating mean annual permafrost table temperature and active layer thickness solely based on temperature monitoring at two depths within the active layer. The approach, based on the TTOP formula, is elegant and could be helpful for the interpretation of field data. The main advantage is that, considering yearly integrated observations of soil temperature at two depths, the proposed method avoids the need of using ground properties measurements, at the price nevertheless of strong simplifying assumptions.

However, the assessment of the performance of the proposed method should be thoroughly improved. The validation based on numerical simulations in idealized cases is not relevant, using an outdated modelling approach as the reference. The validation against field data is good, but too few sites are considered. Once these problems solved, the discussion of the limitations related to the strong underlying assumptions (e.g.: constant ground properties) should be carefully made.

Thus I recommend major revisions of this manuscript prior to consider its publication in The Cryosphere.

**AC:** We are well aware that there exist highly sophisticated models of heat and water transfer in freezing and thawing grounds and that the numerical model used may be viewed as "outdated" or "old-fashioned" from this perspective. However, we still do believe that the numerical simulations for idealized scenarios are relevant because they are meant to evaluate if the analytical–statistical models were derived correctly. Hence, the numerical model is simple so that it can mimic the behaviour of the analytical–statistical models together with their (simplifying) assumptions and boundary conditions, which is a standard practice. We think it does a right (and required) job in this respect because the numerical validations are not obscured by processes that the analytical–statistical models do not account for. Conversely, even sophisticated numerical freeze-thaw models are frequently simplified (e.g., Westermann et al., 2023) so that they can be validated by some benchmark analytical solutions (sinusoidal oscillations or step changes in upper boundary temperature for subsurface temperatures, Neumann or Stefan solutions for freeze-thaw depths) under idealized/simple conditions. This might also be considered simplistic and therefore "outdated" or "old-fashioned", but is not, as the benchmarks are exact solutions for the given problems. We will incorporate some of the above thoughts in the revised manuscript.

Complex scenarios with heat and water transfer in heterogeneous freezing and thawing grounds are where field data have their place in terms of the model validations. We will roughly double the number of the validation sites in the revised manuscript, which will cover much of the world's major permafrost regions (Antarctica, Arctic, Tibetan Plateau and high mountains) and offer us greater heterogeneity in terms of permafrost table temperature and active-layer thickness (approximately −19 °C to −2 °C for MAPT and <50 cm to >200 cm for ALT), as well as in terms of surface cover, active-layer composition and stratigraphy, or permafrost zones.

Westermann, S., et al. (2023). The CryoGrid community model (version 1.0) – a multi-physics toolbox for climate-driven simulations in the terrestrial cryosphere, Geosci. Model Dev., 16, 2607–2647, https://doi.org/10.5194/gmd-16-2607-2023

**RC1:** SPECIFIC COMMENTS

l 8: "which corroborated their theoretical assumptions under idealized scenarios" ; unclear, please rephrase.

**AC:** We will remove this sentence from the abstract because it is unnecessary there.

**RC1:** l 66: "Besides surface temperatures, Eq. (1) is valid for temperatures measured at any depth in the active layer". Please clarify here what is exactly meant by 'is valid' ; has it been validated against field data? With which general procedure?

**AC:** It should mean it works with data from any depth in the active layer, which was suggested in Riseborough (2004).

Riseborough, D. W. (2004). Exploring the parameters of a simple model of the permafrost–climate relationship. PhD Dissertation, Carleton University, Ottawa, 330 pp.

**RC1:** l 76: Eq. (5) (and generally the TTOP formula used here) imply the assumption that the thawed soil thermal conductivity kt is constant over time, both at seasonal and multi-annual time scale. The frozen soil thermal conductivity kf is also considered as constant at multi-annual time scale I guess. Since kt does depends on the soil water content, it varies within an active season and along the years according to the variability of precipitations (and thus infiltration, and thus soil water content). This is a strong assumption that must be pointed out here and extensively discussed in the paper.

**AC:** We agree that the assumption that thermal conductivities of both thawed and frozen ground, and thereby their ratio, are constant at (usually) multi-annual time scales is very simplifying, as ground water content varies within and between years. However, the thermal conductivity ratio expressed by Eq. (5) utilizes thawing and freezing indices at two distinct depths in the active layer. This means that the thermal conductivity ratio involves seasonal variations in ground water content, and thereby thermal conductivity, within the depth layer between temperature sensors used. And since the thawing and freezing indices are calculated annually, the thermal conductivity ratio varies between years. We consider this to be a major improvement over the classic approach with constant thermal properties/thermal conductivity ratio (that were, moreover, frequently only estimated, not measured!). We will discuss this more extensively in the revised manuscript.

**RC1:** l 94-95: "This documents that Eq. (8) for MAPT is analytical and statistical at the same time because it integrates both approaches." ; I don't understand.

**AC:** This should simply state that two different derivations ([1] analytical approach with two TTOP formulas for two distinct depths in the active layer and [2] statistical approach with linear extrapolation based on freezing indices for two distinct depths in the active layer) both lead to the same final formula, which is nice and notable. However, we will try to reformulate this a little in the revised manuscript so that it is more understandable.

**RC1:** l 100: Eq. (13) implies that the volumetric water content ϕ is constant over time, although it varies within an active season and at multi-annual time scale depending on precipitations. Same remark for kt (see also my specific comment at line 76). This is a strong assumption that must be pointed out here and extensively discussed in the paper.

**AC:** The same response as to that at line 76 also applies here. Similarly, we will discuss this more extensively in the revised manuscript.

**RC1:** l 128-129: "Usually, Eq. (21) has been referred to as the modified Stefan model and proved to be useful in situations where the ground physical properties were unavailable and/or for spatial modelling of ALT". Eq. (21) and eq. (13) are strictly equivalent. May be that the difference that the authors want to point out is that the edaphic term in (21) maybe calibrated in itself, without estimating the thawed heat conductivity and the volumetric water content separately. But would it be really different to make a two parameters calibration for kt and ϕ? Anyway these ones would be estimated averages, probably calibrated as well, since these quantities do vary in time (see specific points l 76 and l 100).

**AC:** It is true that Eq. (13) and (21) are equivalent, but they were used separately in previous publications because of their different requirements for input parameters. As you pointed out, Eq. (13) requires thawed thermal conductivity and volumetric water content as two separate variables, which can be determined by field measurements. By contrast, Eq. (21) uses a combination of both in the form of the edaphic term (Eq. 22), which can be calibrated using thawing index and active-layer thickness. Both approaches have their advantages and disadvantages. For us, it is important to show that thermal conductivity and water content can be combined, which

essentially leads to Eq. (23) and (24) that, however, do not require the active-layer thickness to estimate the edaphic factor (…to estimate the active-layer thickness).

**RC1:** l 157-158: "As with Eq. (8), this documents that Eq. (27) for ALT is analytical and statistical at the same time because it integrates both approaches." ; I don't understand.

**AC:** Similar to above, this should simply state that two different derivations ([1] analytical approach with two Stefan formulas for two distinct depths in the active layer and [2] statistical approach with linear extrapolation based on squared thawing indices for two distinct depths in the active layer) both lead to the same final formula, which is nice and notable. However, we will also try to reformulate this a little in the revised manuscript so that it is more understandable.

**RC1:** l 168: The numerical model used for solving heat transfer in the active layer is a very old fashioned one (Carslaw and Jaeger, 1959). Since then numerous modelling works as been done for the simulation of heat and water transfers in soils with freeze-thaw (see for instance the benchmark of Grenier et al., 2018, or the reviews of Bui et al., 2020 and Hu et al., 2023). A more up to date model should be used.

C. Grenier, H. Anbergen, V. Bense, et al., Adv. Water Resour. 114 (2018) 196–218, https://doi.org/10.1016/j.advwatres.2018.02.001

M.T. Bui, J. Lu, L. Nie, A review of hydrological models applied in the permafrost-dominated Arctic region, Geosciences 10 (2020) 401, https://doi.org/10.3390/geosciences10100401

Hu G., Zhao L, Li R., Park H., Wu X., Su Y., Guggenberger G., Wu T., Zou D., Zhu X., Zhang W., Wu Y., Hao J.: Water and heat coupling processes and its simulation in frozen soils: Current status and future research directions, CATENA, Volume 222, 106844, ISSN 0341-8162, doi:10.1016/j.catena.2022.106844, 2023.

**AC:** As argued above, the numerical simulations are meant to evaluate if the analytical–statistical models were derived correctly. Hence, the numerical model is simple so that it can mimic the behaviour of the analytical–statistical models together with their (simplifying) assumptions and boundary conditions, which is a standard practice. We think it does a right (and required) job in this respect because the numerical validations are not obscured by processes that the analytical–statistical models do not account for. Complex scenarios with heat and water transfer in heterogeneous freezing and thawing grounds are where field data have their place in terms of the model validations.

**RC1:** l 179: Using eq. (32) for the reference numerical simulations prevent to consider the effect of the coupling of water flow and heat transfer, since volumetric water content is considered as constant (see table 1). Meanwhile, spatial and temporal variations of water content may be of primary importance for soil thermal regime (see for instance Kurylyk and Watanabe 2013, Sjöberg et al. 2016, Orgogozo et al., 2019). A more complete model should be used.

Kurylyk B.L., Watanabe K., 2013. The mathematical representation of freezing and thawing processes in variably-saturated, non-deformable soils, Advances in Water Resources, Volume 60, Pages 160-177, ISSN 0309-1708, doi:10.1016/j.advwatres.2013.07.016., 2013.

Sjöberg Y., Coon E., Sannel A. B. K., Pannetier R., Harp D., Frampton A., Painter S. L. and Lyon S. W., 2016. Thermal effects of groundwater flow through subarctic fens: A case study based on field observations. doi:10.1002/2015WR017571, 2016.

L. Orgogozo, A.S. Prokushkin, O.S. Pokrovsky, C. Grenier, M. Quintard, J. Viers, S. Audry, Permafr. Periglac. Process. 30 (2019) 75–89, https://doi.org/10.1002/ppp.1995

**AC:** As argued above, we agree that spatial and temporal variations in water content can substantially affect the ground thermal regime. On the other hand, the TTOP and Stefan models, on which the newly devised analytical–

statistical models build, explicitly assume (in their simplest versions) that water content is constant. Since the numerical model is intended to mimic the behaviour of the analytical–statistical models together with their (simplifying) assumptions and boundary conditions, it also considers the water content as constant. We think the numerical model does a right (and required) job in this respect because the numerical validations are not obscured by processes that the analytical–statistical models do not account for. Complex scenarios with heat and water transfer in heterogeneous freezing and thawing grounds are where field data have their place in terms of the model validations.

**RC1:** l 256: "Overall, however, these findings corroborate the theoretical assumptions outlined in Sect. 2.2" Please be more specific. Which precise assumptions?

**AC:** This should mean the modelling approach proved to be feasible based on the numerical tests. We will change this statement in the revised manuscript so that it is clearer.

**RC1:** l 263: "ALT estimates by Eq. (27) were more accurate in Antarctica" I think that this is due to the fact that soil water content varies much more in the Alaskan sites that in the Antarctica sites. The bimodal distribution in the bottom left graph of Figure 5 is maybe also due to this.

**AC:** This is right and we will incorporate it into the revised manuscript. Still, we think this is mainly caused by the active-layer stratigraphy, which is almost exclusively one-layer in Antarctica and two-layer in Alaska. However, this pattern can change because the number of validation sites will roughly double in the revised manuscript.

**RC1:** l 271: "a reasonable accuracy" ; reasonable according to which criterium ?

**AC:** This was meant in the context of previous publications that used the TTOP and Stefan models, but we agree it is vague. This will be changed in the revised manuscript.

**RC1:** l 272: "which corroborated their theoretical assumptions (see Sect. 2.1 and 2.2)" ; unclear, please rephrase.

**AC:** This should mean the modelling approach proved to be feasible based on the numerical tests. We will change this statement in the revise manuscript so that it is clearer.

**RC1:** l 273: "they can work reasonably well under a wide range of climates and ground physical conditions" ; this has to be demonstrated by the discussion.

**AC:** Especially, this was demonstrated by the validation results based on field measurements. However, as stated above, we will roughly double the number of the validation sites in the revised manuscript, which will cover much of the world's major permafrost regions (Antarctica, Arctic, Tibetan Plateau and high mountains) and offer us greater heterogeneity in terms of permafrost table temperature and active-layer thickness (approximately −19 °C to −2 °C for MAPT and <50 cm to >200 cm for ALT), as well as in terms of surface cover, active-layer composition and stratigraphy, or permafrost zones. This will be shown in the results and more extensively discussed in the revised manuscript.

**RC1:** l 277-278: "Under field conditions, the ASM deviations were close to zero on average" ; this statement seems not in line with the results shown in Figure 5.

**AC:** In fact, it cannot even be consistent with Figure 5 because it discusses the mean annual permafrost table temperature, whereas Figure 5 shows the active-layer thickness.

**RC1:** l 315-317: "Since ASMs build solely on thawing and freezing indices at two distinct depths in the active layer, the values of which reflect the rate of heat transfer across their intermediate layer, the solutions also intrinsically account for the temporal variability of ground physical properties." ; I do not agree. According to eq. (1), the TTOP formula on which is based the ASMs does not take into account these temporal variabilities.

**AC:** Yes, the TTOP formula given by Eq. (1) does not indeed account for temporal changes in ground physical properties. However, the ASMs do because they utilize thawing and freezing indices at two distinct depths in the active layer, the relationships (or gradients) between which are determined by natural variability in ground physical properties within the depth layer between temperature sensors throughout the year. This is a fundamental misunderstanding of the whole procedure. Think of it like, for instance, ground thermal diffusivity calculations, which build on damping temperature amplitudes or phase lags between two distinct depths, which is governed by the thermal diffusivity of the ground layer in between. The ASMs have in principle the same rationale. We will try to incorporate some of the above thoughts in the revised manuscript so that this is clearer.

**RC1:** l 317-319: "Likewise, they consider latent and sensible heat and any other factors that might affect the heat transfer in the active layer, some of which other models do not explicitly account for." ; same thing that the previous comment.

**AC:** Same response as the previous comment.

**RC1:** l 325-326: "Ground physical properties also commonly show more or less variability on seasonal and annual time scales […] which most other models cannot handle because they typically treat ground physical properties as constants." ; I think that it is also the case here, according to eq. (1).

**AC:** Same response as the previous comment.

**RC1:** l 359-360: "ASMs for estimating MAPT and ALT can find applications under a wide range of climates and ground physical conditions" ; it sounds to me like an overstatement. Two sites where investigated, largely not enough to sample the variety of permafrost environments: continuous/discontinuous/sporadic, in various environments such as for instance tundra or boreal forest, with diverse lithology and pedology, under various climatic (e.g. precipitation) conditions, etc.

**AC:** Please note we did not investigate only two sites, but two types of permafrost environments (Antarctica and Alaska) where there were a total of 17 sites. As stated above, moreover, we will roughly double the number of the validation sites in the revised manuscript, which will cover much of the world's major permafrost regions (Antarctica, Arctic, Tibetan Plateau and high mountains) and offer us greater heterogeneity in terms of permafrost table temperature and active-layer thickness (approximately −19 °C to −2 °C for MAPT and <50 cm to >200 cm for ALT), as well as in terms of surface cover, active-layer composition and stratigraphy, or permafrost zones. We will then update (or possibly not) the statement based on the results of these more extensive validations.

**RC1:** TECHNICAL CORRECTIONS

l 288-289: "in the order of tenths to first degrees Celsius" ; english language problem.

**AC:** We will change this in the revised manuscript.

**RC1:** l 346-350: Not necessary.

**AC:** We will remove this from the revised manuscript.

---

## Author Comment (AC2)

**AUTHORS' RESPONSE TO THE COMMENTS OF THE REFEREE #2**

**RC2:** The manuscript by Uxa et al. presents an approach for determining MAPT and ALT utilizing shallow ground temperatures at two depths. The MS is interesting and generally well written with a good description of the approach being proposed.

However, there are some concerns regarding the validation of the approach and evidence that it is novel given that the two variables being determined are commonly calculated by interpolation/extrapolation when shallow ground temperatures are available. It is unclear what advantage the method proposed has over this commonly used approach especially given that the authors acknowledge that their equation is in principle a linear extrapolation of the temperature indices. The MS would benefit from a comparison of their model results to those from interpolation/extrapolation of observed shallow ground temperatures. The authors may have done this when they compare their model results to observed MAPT and ALT, but it is not clear how the observed values were determined, and additional explanation is required (see additional comments below). The authors may also want to consult Riseborough (2008, ICOP) regarding use of interpolation and extrapolation to determine thaw depth and importance of spacing of sensor depths.

**AC:** The models are primarily intended to be used for MAPT or ALT estimates at places where ground temperature measurements are too shallow and MAPT or ALT cannot be determined directly or where the spacing between temperature sensors is too large (there are many such stations). Among other possible applications, they can be used to establish typical values of thermal conductivity ratio (for MAPT) and edaphic factor (for ALT), which can then be used to model MAPT and ALT in the past or in the future. Note that the thermal conductivity ratio has never been estimated this way. Edaphic factor has been estimated using the near-surface thawing index and active-layer thickness, but obviously this procedure cannot be used if ground temperatures are too shallow and active-layer thickness is unknown. Our procedure is applicable even in such situations.

The extrapolations have frequently been done rather intuitively and without a physical basis. We admit that the situation is somewhat better in the case of the active-layer thickness, but in principle none has been done for extrapolating the permafrost table temperature. The linear relationship between the thawing and freezing indices within the active layer is new, as is the whole procedure, which has a physical basis. We will emphasize these thoughts a bit more in the revised manuscript.

The ALT values were determined by continuous tracking/interpolating the 0 °C isotherm from measured temperatures (see e.g., Hrbáček et al., 2020, 2021; Kňažková and Hrbáček, 2024). The mean annual permafrost table temperature (MAPT) was the mean annual ground temperature interpolated to the depth corresponding to ALT (=permafrost table). We compared our model outputs directly with these measured/interpolated values. Thanks for providing the reference to Riseborough (2008). If it proves useful, we will consider including it in the revised manuscript.

Hrbáček, F., Cannone, N., Kňažková, M., Malfasi, F., Convey, P., Guglielmin, M. (2020). Effect of climate and moss vegetation on ground surface temperature and the active layer among different biogeographical regions in Antarctica. Catena, 190, 104562. https://doi.org/10.1016/j.catena.2020.104562

Hrbáček, F., Engel, Z., Kňažková, M., Smolíková, J. (2021). Effect of summer snow cover on the active layer thermal regime and thickness on CALM-S JGM site, James Ross Island, eastern Antarctic Peninsula. Catena, 207, 105608. https://doi.org/10.1016/j.catena.2021.105608

Kňažková, M., Hrbáček, F. (2024). Interannual variability of soil thermal conductivity and moisture on the Abernethy Flats (James Ross Island) during thawing seasons 2015–2023. Catena, 234, 107640. https://doi.org/10.1016/j.catena.2023.107640

**RC2:** The analysis and conclusions would benefit from better descriptions of the field sites including material properties, vegetation, climate etc. The limited range in their characteristics is a concern as all the sites are in cold permafrost of the continuous zone and likely tundra sites. This limits the conclusions that can be made regarding

model performance and the broader applicability of the approach. Consideration of sites in warmer permafrost in the discontinuous zone including those in forested and peatland terrain would be useful as this would further back up statements regarding model performance including statements made regarding warmer permafrost.

**AC:** We will roughly double the number of the validation sites in the revised manuscript, which will cover much of the world's major permafrost regions (Antarctica, Arctic, Tibetan Plateau and high mountains) and offer us greater heterogeneity in terms of permafrost table temperature and active-layer thickness (approximately −19 °C to −2 °C for MAPT and <50 cm to >200 cm for ALT), as well as in terms of surface cover, active-layer composition and stratigraphy, or permafrost zones. We will also provide more key information on the validation sites in the revised manuscript as suggested. This will allow us to make the validations more robust.

**RC2:** Additional comments related to the concerns raised above and other comments are provided below for the authors' consideration.

**RC2:** L21 – "indicators" might be better than "measures"

**AC:** We will change it as suggested.

**RC2:** L22-23 – suggested revision: "Climate change has resulted in permafrost warming and active-layer thickening throughout the permafrost regions". Biskaborn et al. is now out of date with respect to the trends. I suggest you include Smith et al. (2024, State of Climate- Arctic), along with Noetzli et al. (2024), as it provides the details for Arctic and is up to date.

**AC:** We will change it as suggested.

**RC2:** L29-30 – It is important to note that active layer thickness is not determined directly from geophysical surveys but is interpreted and it is difficult to determine ALT in warm permafrost with high unfrozen water contents.

**AC:** We totally agree and that is why we only state that geophysical surveys are among the methods used to investigate permafrost and active-layer thickness. We did not mention anywhere in the original manuscript that active-layer thickness is determined directly using geophysics.

**RC2:** L27-34 – Smith and Brown (2009) outline the various methods used as does Streletskiy et al. (2022).

**AC:** Thank you for these useful references. We will incorporate them into the revised manuscript.

**RC2:** L34-37 – Even if ground temperatures are measured within shallow permafrost and the active layer the permafrost table temperature or active layer thickness still needs to be determined/calculated. Interpolation and extrapolation is the method usually used. I suggest you consult Riseborough (2008 ICOP) which describes the appropriateness of interpolation/extrapolation approaches.

**AC:** Thank you for this reference. If it proves useful, we will consider including it in the revised manuscript.

**RC2:** L38-42 – The purpose of the model is important. The ones mentioned are generally used for predictive applications such as determining conditions with little information on the site conditions. What you seem to be proposing is away to determining ALT or MAPT based on having ground temperature measurements which is what we do when we use interpolation/extrapolation of shallow ground temperatures to determine ALT or MAPT (see comment above).

**AC:** The models are primarily intended to be used for MAPT or ALT estimates at places where ground temperature measurements are too shallow and MAPT or ALT cannot be determined directly or where the spacing between temperature sensors is too large (there are many such stations). Among other possible applications, they can be used to establish typical values of thermal conductivity ratio (for MAPT) and edaphic factor (for ALT), which can then be used to model MAPT and ALT in the past or in the future. Note that the thermal conductivity ratio has never

been estimated this way. Edaphic factor has been estimated using the near-surface thawing index and active-layer thickness, but obviously this procedure cannot be used if ground temperatures are too shallow and active-layer thickness is unknown. Our procedure is applicable even in such situations. We will emphasize the above thoughts in the revised manuscript.

**RC2:** L45 – Is reference being made to air or ground temperatures here?

**AC:** It was meant to be both, because both air and ground temperatures are used to as model forcings (of course, for air temperatures, some procedures must be used to convert them to ground temperatures). We will explicitly state this in the revised manuscript.

**RC2:** L46-49 – Some of the approaches mentioned do consider variable properties including thermal conductivity for thawed and frozen conditions (e.g. ratio between them is included in TTOP equation) or the variation of conductivity with temperature (and unfrozen water content).

**AC:** Yes, some (but few) of the approaches mentioned indeed considered the temporal variations in ground physical properties, but these variations would have to be involved in the models as additional forcings, which is frequently not available. Hence, most models treat the ground physical properties as constants, which brings complications in terms of their representativeness (for instance, one measurement per year or more should represent the temporal variations over this whole period). We believe that our solutions are useful in that they can simply address this general lack and/or non-representativeness of ground physical data. We will a bit revise this sentence in the revised manuscript so that the above thoughts are clearer.

**RC2:** L57 – Editorial suggestion – delete "Besides other solution (Garagulya, 1990)" – it is not adding anything.

**AC:** We will remove it from the revised manuscript.

**RC2:** L66 – Editorial suggestion – Delete first part of sentence: "Eq. (1) is also valid for temperatures measured…..layer, which is convenient because…."

**AC:** We will change it in the revised manuscript as suggested.

**RC2:** L68 – Note that usually the reference to surface temperature measurements used in these equations is from a sensor in upper 3-5 cm of the ground so not using exact surface temperature. The surface temperature can be estimated using n-factor to provide input into the equation.

**AC:** We agree and will consider this in the revised manuscript. However, the problems associated with ground surface temperature measurements (mentioned in the original manuscript) also relate to some extent to near-surface measurements.

**RC2:** L84 – If this is essentially extrapolation how is this different from the approach others use to determine the depth of the permafrost table and MAPT when they have temperatures at two or more depths?

**AC:** As detailed above, the major difference is that previous extrapolations have frequently been done rather intuitively and without a physical basis, which is not the case of our models. We will emphasize these thoughts a bit more in the revised manuscript.

**RC2:** L97 – Editorial suggestion - Delete first part of sentence: "ALT (m) can be calculated using the….

**AC:** This will require more changes to the text than this to make it understandable, but we will follow this suggestion in the revised manuscript.

**RC2:** L99 – missing word: "…simplest form is as…."

**AC:** We will add this to the revised manuscript.

**RC2:** L103 – See earlier comment regarding estimates of surface temperature used by others.

**AC:** We agree and will consider this in the revised manuscript. However, the problems associated with ground surface temperature measurements (mentioned in the original manuscript) also relate to some extent to near-surface measurements.

**RC2:** L136 – Smith et al. (2009) is also relevant - used observed ALT from sites in various regions and environments (tundra, forest, peatland and mineral and organic soil) to show the range in the Edaphic factor.

**AC:** Thank you for this useful reference. We will incorporate it in the revised manuscript.

**RC2:** L148 – See earlier comment regarding extrapolation

**AC:** The major difference is that previous extrapolations were frequently done rather intuitively and without a physical basis. We will emphasize these thoughts a bit more in the revised manuscript.

**RC2:** L159-197 – It seems that there are several assumptions being made as well as simplifications. For example, thermal conductivity changes with temperature due to change in unfrozen water but it appears that constant frozen and unfrozen conductivity are assumed. Are the results of the two models really comparable?

**AC:** Yes, there are several assumptions and simplifications that needed to be made so that the numerical model can mimic the behaviour of the analytical–statistical models together with their (simplifying) assumptions and boundary conditions, which is a standard practice. We think the numerical model does a right (and required) job in this respect because the numerical validations are not obscured by processes that the analytical–statistical models do not account for. We believe that the numerical simulations for idealized scenarios are relevant because they are meant to evaluate if the analytical–statistical models were derived correctly.

**RC2:** Table 2 – For years and seasons is the 2nd number the total record length and the first number the number of years utilized in analysis? It might be clearer to refer to number of years used and refer to it as e.g. 6 of 6. Are the ALT values determined from temperature or through probing? How is MAPT determined? It is unclear what you are comparing your modelled values with? I might have missed something here but maybe there needs to be a clearer explanation

**AC:** We used three depth combinations in our models based on temperatures from the depth intervals of 0–10 cm, 25–35 cm and 45–55 cm. However, there were occasional gaps in the datasets, which caused that some of the depth combinations could not sometimes be used for validations or that the number of years for MAPT and ALT validations differed. We admit this is not very intuitive, and we will rework it to make it fully understandable in the revised manuscript.

The ALT values were determined by continuous tracking/interpolating the 0 °C isotherm from measured temperatures (see e.g., Hrbáček et al., 2020, 2021; Kňažková and Hrbáček, 2024). The mean annual permafrost table temperature (MAPT) was the mean annual ground temperature interpolated to the depth corresponding to ALT (=permafrost table). We compared our model outputs directly with these measured/interpolated values.

**RC2:** L200-201 – No information on the sites is provided so the reader doesn't know how diverse they are. There is no information provided on material characteristics or vegetation. The Alaskan sites are all on the North Slope in the continuous permafrost zone and likely in tundra environments, so conditions are not that diverse with respect to climate and vegetation. Using field data from sites in warmer permafrost in discontinuous zone and for forested sites would provide more diverse conditions. This would help show if your approach is valid for a wide range in conditions.

**AC:** We will roughly double the number of the validation sites in the revised manuscript, which will cover much of the world's major permafrost regions (Antarctica, Arctic, Tibetan Plateau and high mountains) and offer us greater heterogeneity in terms of permafrost table temperature and active-layer thickness (approximately −19 °C to −2 °C for MAPT and <50 cm to >200 cm for ALT), as well as in terms of surface cover, active-layer composition and stratigraphy, or permafrost zones. We will also provide general information on the validation sites in the revised manuscript as suggested.

**RC2:** L204 – Since you are referring to a depth it would be better to refer to permafrost table or base of the active layer. Do you mean the base of the active layer was above the shallowest sensor?

**AC:** We will revise this sentence in the revised manuscript so that it is clearer. Please note that we used temperatures measured at the depth intervals of 0–10 cm, 25–35 cm and 45–55 cm as model forcings so that they are comparable across the study sites, but the depths at each site were constant, for instance, 5 cm, 30 cm and 50 cm. If the active layer is thin, there can be some years when the base of the active layer was shallower than the deepest sensor used. This means that the active-layer thickness was less than 50 cm. But definitely, we did not mention anywhere that the base of the active layer was above the shallowest temperature sensor.

**RC2:** L206-2011 – See Riseborough et al. (2008 ICOP) regarding errors associated with different approaches (interpolation, extrapolation) to determine thaw depth/top of permafrost etc. and guidance on the best approach to use.

**AC:** Thank you for this reference. If it proves useful, we will consider including it in the revised manuscript. Please note that we used linear interpolation based on measured temperatures to determined MAPT and ALT, which we also used in numerous previous publications.

**RC2:** L222 – Wasn't this exponential decrease already fairly well known? Doesn't the magnitude of the decrease depend on the material properties?

**AC:** It was known for thawing indices (e.g., Riseborough, 2003), but rather neglected for freezing indices because permafrost studies have dominantly focused on summer active-layer dynamics and/or annual means. However, the linear relationship between the thawing and freezing indices within the active layer is new. And yes, the magnitude of the decrease mostly depends on the thermal conductivities and the amount of latent heat. We will emphasize these points a bit more in the revised manuscript.

Riseborough, D. (2003). Thawing and freezing indices in the active layer, in: Proceedings of the 8th International Conference on Permafrost, Zurich, Switzerland, 21–25 July 2003, 953-–958, 2003.

**RC2:** L235 – See earlier comment – How were observed values determined?

**AC:** The observed MAPT values were determined by a linear interpolation of the mean annual ground temperatures observed at sensors just above and below the observed active-layer thickness, which was briefly described in Sect. 3.3. Still, we will make this a bit clearer in the revised manuscript so that it is easily understandable.

**RC2:** L260 – See earlier comment – How were observed values determined?

**AC:** The ALT values were determined by continuous tracking/interpolating the 0 °C isotherm from measured temperatures (see e.g., Hrbáček et al., 2020, 2021; Kňažková and Hrbáček, 2024), which was briefly described in Sect. 3.3. Still, we will make this a bit clearer in the revised manuscript so that it is easily understandable.

**RC2:** L265-268 – Does the difference in error between Antarctica and Alaska sites have anything to do with the material properties. Was latent heat more of a factor for the AK sites?

**AC:** We believe that it does. We think this is mainly caused by the active-layer stratigraphy, which is almost exclusively one-layer without any organic material in Antarctica and two-layer in Alaska. Hence, it definitely also has something to do with larger amount of latent heat at the Alaskan sites.

**RC2:** L264-292 – The thermal offset depends on the ratio of thawed and frozen thermal conductivity which depends on the amount of moisture/ice in the ground. If the moisture content is low or arid conditions exist, then the offset will be low or positive. Is the site in McMurdo Sound a dry site? It would be useful to know this. It would have been good to use sites with warmer permafrost in your analysis to back up the comment that deviation in MAPT estimates would be larger.

**AC:** Yes, the sites in McMurdo Sound experience hyperarid conditions, which most likely produce positive thermal offsets. Please note that we will roughly double the number of the validation sites in the revised manuscript, which will cover much of the world's major permafrost regions (Antarctica, Arctic, Tibetan Plateau and high mountains) and offer us greater heterogeneity in terms of permafrost table temperature and active-layer thickness (approximately −19 °C to −2 °C for MAPT and <50 cm to >200 cm for ALT), as well as in terms of surface cover, active-layer composition and stratigraphy, or permafrost zones. Hence, we will be able to back up or revise our statement that larger deviations in MAPT would be expected in warmer conditions.

**RC2:** L306-310 – It would be useful to have information on the material properties at the field sites to back up these statements.

**AC:** We totally agree that it would be useful, but unfortunately the information on the material properties are scattered or rather general/descriptive from the validation sites. Hence, we can draw only general conclusions in this respect. If we wanted to have detailed information on the material properties, the number of validation sites would have to be much smaller, which is, however, undesirable in terms of robustness of the validations.

**RC2:** L315-330 – Although these other approaches make assumptions regarding thermal properties etc. based on general site characteristics, information on ground temperature is not required and the models determine the ground temperatures. This makes them useful for determining current and future conditions. This might make them more broadly applicable.

**AC:** Unfortunately, we do not understand this comment clearly. Anyway, it is important to note that all other models for permafrost table temperature and active-layer thickness require information on ground (surface or near-surface) temperature, which is used as their upper boundary condition.

**RC2:** L327-330 – If temperature below the permafrost table was available would it be used if there were only one sensor at a shallower depth? You state that inputs can be any depth combination within the active layer based on temperature data availability and site characteristics. What are the site characteristics being referred to?

**AC:** Unfortunately, temperatures below the permafrost table cannot be used because thawing indices must have non-zero (=positive) values; otherwise the outputs would be erroneous. We will explicitly state this in the revised manuscript.
By the site characteristics we meant that specific depth combinations may work better under specific site characteristics. However, we acknowledge that this is unclear and rather misleading, and therefore we will remove it from the revised manuscript.

**RC2:** L331-334 – Aren't these products based on modelling with various assumptions made regarding ground properties etc.

**AC:** Yes, but these assumptions and/or ground properties are largely unknown for these products, which considerably impedes model applications. Consequently, for instance, the active-layer thickness estimates are limited by the deepest ground temperature level (node) available in these products, which is frequently shallow and situated within the active layer. However, our models can deal with such situations. We will emphasize this a bit further in the revised manuscript.

**RC2:** L338-340 – This is likely one of the primary sources of error especially with respect to moisture/ice contents and latent heat effects as discussed in Riseborough (2003).

**AC:** Yes, it certainly is, as in any analytical model, and we think it is only fair to admit it.

**RC2:** L340-348 – Riseborough (2008) is probably relevant here especially with respect to spacing of temperature measurements etc. in determining thaw depth.

**AC:** We think this is much more relevant, for instance, to interpolating the active-layer thickness or calculating the permafrost table temperature from measured ground temperatures. In terms of this paper, this is therefore particularly relevant to the validation data used to evaluate the models. However, the models themselves are in principle independent of the spacing of temperature measurements. More important is the sensor position with respect to the active-layer stratigraphy.

**RC2:** References

Riseborough, D.W. 2008. Estimating active layer and talik thickness from temperature data: implications from modeling results. In Ninth International Conference on Permafrost. Edited by D.L. Kane and K.M. Hinkel. Fairbanks, Alaska. Institute of Northern Engineering, University of Alaska Fairbanks, Vol.2, pp. 1487-1492.

Smith, S.L., Romanovsky, V.E., Isaksen, K., Nyland, K., Shiklomanov, N.I., Streletskiy, D.A., and Christiansen, H.H. 2024. Permafrost (Arctic) [in "State of the Climate in 2023"]. Bulletin of the American Meteorological Society (supplement), 105(8): S314-S317. doi:10.1175/BAMS-D-24-0101.1

Smith, S.L., Wolfe, S.A., Riseborough, D.W., and Nixon, F.M. 2009. Active-layer characteristics and summer climatic indices, Mackenzie Valley, Northwest Territories, Canada. Permafrost and Periglacial Processes, 20(2): 201-220. doi:10.1002/ppp.651

Smith, S. and Brown, J. 2009: Assessment of the status of the development of the standards for the Terrestrial Essential Climate Variables - T7 - Permafrost and seasonally frozen ground. GTOS 62 Essential Climate Variables

Streletskiy et al. 2022 Measurement Recommendations and Guidelines for the Global Terrestrial Network for Permafrost (GTN-P). DOI: 10.5281/zenodo.5973079

---

## Author Response (AR1)

**AUTHORS' RESPONSE TO THE COMMENTS OF THE EDITOR**

Dear Tomáš Uxa and co-authors,

Thank you for submitting your answers to the two review reports and for sketching the principal course of action your foresee. The main comments relate to the validation or performance measures of your approach. In this regard, you primarily suggest to increase the number of field sites across the globe. This is certainly valuable. Considering the criticism of reviewer #1 on the comparison to an 'outdated approach', I do agree. In principal, you mainly check the validity of the numerical implementation giving you some comfort in your analytical implementation. This is important but readers would rather wonder about the limitations of your analytical approach and therefore a comparison to a contemporatry model that includes physical process beyond what you consider. Such a comparison might also help you to vindicate your approach with regard to temporarily varying parameters. Moreover, reviewer #2 points you to literature for other extrapolation approaches. Please consider relevant techniques for additional comparison at the field sites or synthetic experiments.

I urge you to honestly expand your comparison either with regard to state-of-the-art modelling or to other extrapolation techniques. In this way, the applicability and limitations of your approach can easier be assessed.

In summary, I invite you to submit a revised version of your manuscript, which will undergo a second round of reviews.

The editor, Johannes Fürst

Dear Editor,

On behalf of co-authors, I am submitting a revised version of the manuscript ID EGUSPHERE-2024-2989 entitled "Simple analytical–statistical models (ASMs) for mean annual permafrost table temperature and active-layer thickness estimates" by Tomáš Uxa, Filip Hrbáček, and Michaela Kňažková.

We responded point-by-point to all the reviewers' comments and suggestions and made corresponding revisions in the manuscript. Our responses are included at the end of this text, starting on page 2. Please note that the reviewers' comments are prefixed by bold RC (1 or 2), while the authors' comments are prefixed by bold AC. All the authors' responses are also in blue in order to distinguish them from the reviewers' comments.

The major change in the manuscript is that we decided to completely remove the numerical simulations for idealized scenarios from the revised manuscript, and the model evaluation now fully relies on the field observations. Complex scenarios with heat and water transfer in heterogeneous freezing and thawing grounds are where field data have their place in terms of the model validations.

We more than doubled the number of the validation sites in the revised manuscript, which now cover much of the Earth's major permafrost regions (Antarctica, Arctic, Tibetan Plateau and high mountains) and offer us greater heterogeneity in terms of permafrost table temperature and active-layer thickness (approximately −19 °C to −0 °C for MAPT and ~40 cm to ~300 cm for ALT), as well as in terms of surface cover, active-layer composition and stratigraphy, or permafrost zones. This makes the validations more robust. The summary table listing the key characteristics of the validation sites is provided in the appendix.

We believe that the improvements made in the revised manuscript are sufficient.

Thank for your editorial work on the manuscript, your comments on it, and especially for your patience while awaiting the revised manuscript, which I highly appreciate!

Thank you very much for considering the revised manuscript.

Yours sincerely,

Tomáš Uxa

**AUTHORS' RESPONSE TO THE COMMENTS OF THE REFEREE #1**

**RC1:** GENERAL COMMENTS

The authors propose a simple method for estimating mean annual permafrost table temperature and active layer thickness solely based on temperature monitoring at two depths within the active layer. The approach, based on the TTOP formula, is elegant and could be helpful for the interpretation of field data. The main advantage is that, considering yearly integrated observations of soil temperature at two depths, the proposed method avoids the need of using ground properties measurements, at the price nevertheless of strong simplifying assumptions.

However, the assessment of the performance of the proposed method should be thoroughly improved. The validation based on numerical simulations in idealized cases is not relevant, using an outdated modelling approach as the reference. The validation against field data is good, but too few sites are considered. Once these problems solved, the discussion of the limitations related to the strong underlying assumptions (e.g.: constant ground properties) should be carefully made.

Thus I recommend major revisions of this manuscript prior to consider its publication in The Cryosphere.

**AC:** Please note that we decided to completely remove the numerical simulations for idealized scenarios from the revised manuscript, and the model evaluation now fully relies on the field observations. Complex scenarios with heat and water transfer in heterogeneous freezing and thawing grounds are where field data have their place in terms of the model validations.

We more than doubled the number of the validation sites in the revised manuscript, which now cover much of the Earth's major permafrost regions (Antarctica, Arctic, Tibetan Plateau and high mountains) and offer us greater heterogeneity in terms of permafrost table temperature and active-layer thickness (approximately −19 °C to −0 °C for MAPT and ~40 cm to ~300 cm for ALT), as well as in terms of surface cover, active-layer composition and stratigraphy, or permafrost zones. This makes the validations more robust. The summary table listing the key characteristics of the validation sites is provided in the appendix.

**RC1:** SPECIFIC COMMENTS

l 8: "which corroborated their theoretical assumptions under idealized scenarios" ; unclear, please rephrase.

**AC:** Removed from the revised manuscript.

**RC1:** l 66: "Besides surface temperatures, Eq. (1) is valid for temperatures measured at any depth in the active layer". Please clarify here what is exactly meant by 'is valid' ; has it been validated against field data? With which general procedure?

**AC:** This should mean it works with data from any depth in the active layer, which was suggested in Riseborough (2004).

Riseborough, D. W. (2004). Exploring the parameters of a simple model of the permafrost–climate relationship. PhD Dissertation, Carleton University, Ottawa, 330 pp.

**RC1:** l 76: Eq. (5) (and generally the TTOP formula used here) imply the assumption that the thawed soil thermal conductivity kt is constant over time, both at seasonal and multi-annual time scale. The frozen soil thermal conductivity kf is also considered as constant at multi-annual time scale I guess. Since kt does depends on the soil water content, it varies within an active season and along the years according to the variability of precipitations (and thus infiltration, and thus soil water content). This is a strong assumption that must be pointed out here and extensively discussed in the paper.

**AC:** We agree that the assumption that thermal conductivities of both thawed and frozen ground, and thereby their ratio, are constant at (usually) multi-annual time scales is very simplifying, as ground water content varies within and between years. However, the thermal conductivity ratio expressed by Eq. (5) utilizes thawing and freezing

indices at two distinct depths in the active layer. This means that the thermal conductivity ratio involves seasonal variations in ground water content, and thereby thermal conductivity, within the depth layer between temperature sensors used. And since the thawing and freezing indices are calculated annually, the thermal conductivity ratio varies between years. We consider this to be a major improvement over the classic approach with constant thermal properties/thermal conductivity ratio (that were, moreover, frequently only estimated, not measured). We discuss this more in the revised manuscript.

**RC1:** l 94-95: "This documents that Eq. (8) for MAPT is analytical and statistical at the same time because it integrates both approaches." ; I don't understand.

**AC:** This should simply state that two different derivations ([1] analytical approach with two TTOP formulas for two distinct depths in the active layer and [2] statistical approach with linear extrapolation based on freezing indices for two distinct depths in the active layer) both lead to the same final formula, which is nice and notable. However, we reformulated this a little in the revised manuscript so that it is more understandable.

**RC1:** l 100: Eq. (13) implies that the volumetric water content $\phi$ is constant over time, although it varies within an active season and at multi-annual time scale depending on precipitations. Same remark for kt (see also my specific comment at line 76). This is a strong assumption that must be pointed out here and extensively discussed in the paper.

**AC:** The same response as to that at line 76 also applies here. Similarly, we discuss this more extensively in the revised manuscript.

**RC1:** l 128-129: "Usually, Eq. (21) has been referred to as the modified Stefan model and proved to be useful in situations where the ground physical properties were unavailable and/or for spatial modelling of ALT". Eq. (21) and eq. (13) are strictly equivalent. May be that the difference that the authors want to point out is that the edaphic term in (21) maybe calibrated in itself, without estimating the thawed heat conductivity and the volumetric water content separately. But would it be really different to make a two parameters calibration for kt and $\phi$? Anyway these ones would be estimated averages, probably calibrated as well, since these quantities do vary in time (see specific points l 76 and l 100).

**AC:** It is true that Eq. (13) and (21) are equivalent, but they were used separately in previous publications because of their different requirements for input parameters. As you pointed out, Eq. (13) requires thawed thermal conductivity and volumetric water content as two separate variables, which can be determined by field measurements. By contrast, Eq. (21) uses a combination of both in the form of the edaphic term (Eq. 22), which can be calibrated using thawing index and active-layer thickness. Both approaches have their advantages and disadvantages. For us, it is important to show that thermal conductivity and water content can be combined, which essentially leads to Eq. (23) and (24) that, however, do not require the active-layer thickness to estimate the edaphic factor (…to estimate the active-layer thickness).

**RC1:** l 157-158: "As with Eq. (8), this documents that Eq. (27) for ALT is analytical and statistical at the same time because it integrates both approaches." ; I don't understand.

**AC:** Similar to above, this should simply state that two different derivations ([1] analytical approach with two Stefan formulas for two distinct depths in the active layer and [2] statistical approach with linear extrapolation based on squared thawing indices for two distinct depths in the active layer) both lead to the same final formula, which is nice and notable. However, we reformulated this a little in the revised manuscript so that it is more understandable.

**RC1:** l 168: The numerical model used for solving heat transfer in the active layer is a very old fashioned one (Carslaw and Jaeger, 1959). Since then numerous modelling works as been done for the simulation of heat and water transfers in soils with freeze-thaw (see for instance the benchmark of Grenier et al., 2018, or the reviews of Bui et al., 2020 and Hu et al., 2023). A more up to date model should be used.

C. Grenier, H. Anbergen, V. Bense, et al., Adv. Water Resour. 114 (2018) 196–218, https://doi.org/10.1016/j.advwatres.2018.02.001

M.T. Bui, J. Lu, L. Nie, A review of hydrological models applied in the permafrost-dominated Arctic region, Geosciences 10 (2020) 401, https://doi.org/10.3390/geosciences10100401

Hu G., Zhao L, Li R., Park H., Wu X., Su Y., Guggenberger G., Wu T., Zou D., Zhu X., Zhang W., Wu Y., Hao J.: Water and heat coupling processes and its simulation in frozen soils: Current status and future research directions, CATENA, Volume 222, 106844, ISSN 0341-8162, doi:10.1016/j.catena.2022.106844, 2023.

**AC:** As argued above, we decided to completely remove the numerical simulations for idealized scenarios from the revised manuscript, and the model evaluation now fully relies on the field observations. Complex scenarios with heat and water transfer in heterogeneous freezing and thawing grounds are where field data have their place in terms of the model validations.

**RC1:** l 179: Using eq. (32) for the reference numerical simulations prevent to consider the effect of the coupling of water flow and heat transfer, since volumetric water content is considered as constant (see table 1). Meanwhile, spatial and temporal variations of water content may be of primary importance for soil thermal regime (see for instance Kurylyk and Watanabe 2013, Sjöberg et al. 2016, Orgogozo et al., 2019). A more complete model should be used.

Kurylyk B.L., Watanabe K., 2013. The mathematical representation of freezing and thawing processes in variably-saturated, non-deformable soils, Advances in Water Resources, Volume 60, Pages 160-177, ISSN 0309-1708, doi:10.1016/j.advwatres.2013.07.016., 2013.

Sjöberg Y., Coon E., Sannel A. B. K., Pannetier R., Harp D., Frampton A., Painter S. L. and Lyon S. W., 2016. Thermal effects of groundwater flow through subarctic fens: A case study based on field observations. doi:10.1002/2015WR017571, 2016.

L. Orgogozo, A.S. Prokushkin, O.S. Pokrovsky, C. Grenier, M. Quintard, J. Viers, S. Audry, Permafr. Periglac. Process. 30 (2019) 75–89, https://doi.org/10.1002/ppp.1995

**AC:** As argued above, we decided to completely remove the numerical simulations for idealized scenarios from the revised manuscript, and the model evaluation now fully relies on the field observations. Complex scenarios with heat and water transfer in heterogeneous freezing and thawing grounds are where field data have their place in terms of the model validations.

**RC1:** l 256: "Overall, however, these findings corroborate the theoretical assumptions outlined in Sect. 2.2" Please be more specific. Which precise assumptions?

**AC:** Removed from the revised manuscript.

**RC1:** l 263: "ALT estimates by Eq. (27) were more accurate in Antarctica" I think that this is due to the fact that soil water content varies much more in the Alaskan sites that in the Antarctica sites. The bimodal distribution in the bottom left graph of Figure 5 is maybe also due to this.

**AC:** This is right and we consider it in the revised manuscript. Still, we think this was mainly caused by the active-layer stratigraphy, which is almost exclusively one-layer without any organic material at bare-ground sites, while it is frequently two-layer at vegetated sites.

**RC1:** l 271: "a reasonable accuracy" ; reasonable according to which criterium ?

**AC:** This was meant in the context of previous publications that used the TTOP and Stefan models, but we agree it is vague. Removed from the revised manuscript.

**RC1:** l 272: "which corroborated their theoretical assumptions (see Sect. 2.1 and 2.2)" ; unclear, please rephrase.

**AC:** Removed from the revised manuscript.

**RC1:** l 273: "they can work reasonably well under a wide range of climates and ground physical conditions" ; this has to be demonstrated by the discussion.

**AC:** Especially, this was demonstrated by the validation results based on field measurements. However, as stated above, we more than doubled the number of the validation sites in the revised manuscript, which now cover much of the Earth's major permafrost regions (Antarctica, Arctic, Tibetan Plateau and high mountains) and offer us greater heterogeneity in terms of permafrost table temperature and active-layer thickness (approximately −19 °C to −0 °C for MAPT and ~40 cm to ~300 cm for ALT), as well as in terms of surface cover, active-layer composition and stratigraphy, or permafrost zones. This will be shown in the results and more extensively discussed in the revised manuscript.

**RC1:** l 277-278: "Under field conditions, the ASM deviations were close to zero on average" ; this statement seems not in line with the results shown in Figure 5.

**AC:** In fact, it could not even be consistent with Figure 5 because it discusses the mean annual permafrost table temperature, whereas Figure 5 shows the active-layer thickness.

**RC1:** l 315-317: "Since ASMs build solely on thawing and freezing indices at two distinct depths in the active layer, the values of which reflect the rate of heat transfer across their intermediate layer, the solutions also intrinsically account for the temporal variability of ground physical properties." ; I do not agree. According to eq. (1), the TTOP formula on which is based the ASMs does not take into account these temporal variabilities.

**AC:** Yes, the TTOP formula given by Eq. (1) does not indeed account for temporal changes in ground physical properties. However, the ASMs do because they utilize thawing and freezing indices at two distinct depths in the active layer, the relationships (or gradients) between which are determined by natural variability in ground physical properties within the depth layer between temperature sensors throughout the year. This is a fundamental misunderstanding of the whole procedure. Think of it like, for instance, ground thermal diffusivity calculations, which build on damping temperature amplitudes or phase lags between two distinct depths, which is governed by the thermal diffusivity of the ground layer in between. The ASMs have in principle the same rationale. We incorporated some of the above thoughts in the revised manuscript so that this is clearer.

**RC1:** l 317-319: "Likewise, they consider latent and sensible heat and any other factors that might affect the heat transfer in the active layer, some of which other models do not explicitly account for." ; same thing that the previous comment.

**AC:** Same response as the previous comment.

**RC1:** l 325-326: "Ground physical properties also commonly show more or less variability on seasonal and annual time scales […] which most other models cannot handle because they typically treat ground physical properties as constants." ; I think that it is also the case here, according to eq. (1).

**AC:** Same response as the previous comment.

**RC1:** l 359-360: "ASMs for estimating MAPT and ALT can find applications under a wide range of climates and ground physical conditions" ; it sounds to me like an overstatement. Two sites where investigated, largely not enough to sample the variety of permafrost environments: continuous/discontinuous/sporadic, in various environments such

as for instance tundra or boreal forest, with diverse lithology and pedology, under various climatic (e.g. precipitation) conditions, etc.

**AC:** Please note we did not investigate only two sites, but two types of permafrost environments (Antarctica and Alaska) where there were a total of 17 sites. As stated above, moreover, we more than doubled the number of the validation sites in the revised manuscript, which now cover much of the Earth's major permafrost regions (Antarctica, Arctic, Tibetan Plateau and high mountains) and offer us greater heterogeneity in terms of permafrost table temperature and active-layer thickness (approximately −19 °C to −0 °C for MAPT and ~40 cm to ~300 cm for ALT), as well as in terms of surface cover, active-layer composition and stratigraphy, or permafrost zones.

**RC1:** TECHNICAL CORRECTIONS

l 288-289: "in the order of tenths to first degrees Celsius" ;  english language problem.

**AC:** Removed from the revised manuscript.

**RC1:** l 346-350: Not necessary.

**AC:** Removed from the revised manuscript.

**AUTHORS' RESPONSE TO THE COMMENTS OF THE REFEREE #2**

**RC2:** The manuscript by Uxa et al. presents an approach for determining MAPT and ALT utilizing shallow ground temperatures at two depths. The MS is interesting and generally well written with a good description of the approach being proposed.

However, there are some concerns regarding the validation of the approach and evidence that it is novel given that the two variables being determined are commonly calculated by interpolation/extrapolation when shallow ground temperatures are available. It is unclear what advantage the method proposed has over this commonly used approach especially given that the authors acknowledge that their equation is in principle a linear extrapolation of the temperature indices. The MS would benefit from a comparison of their model results to those from interpolation/extrapolation of observed shallow ground temperatures. The authors may have done this when they compare their model results to observed MAPT and ALT, but it is not clear how the observed values were determined, and additional explanation is required (see additional comments below). The authors may also want to consult Riseborough (2008, ICOP) regarding use of interpolation and extrapolation to determine thaw depth and importance of spacing of sensor depths.

**AC:** The models are primarily intended to be used for MAPT or ALT estimates at places where ground temperature measurements are too shallow and MAPT or ALT cannot be determined directly or where the spacing between temperature sensors is too large (there are many such stations). Among other possible applications, they can be used to establish typical values of thermal conductivity ratio (for MAPT) and edaphic factor (for ALT), which can then be used to model MAPT and ALT in the past or in the future. Note that the thermal conductivity ratio has never been estimated this way. Edaphic factor has been estimated using the near-surface thawing index and active-layer thickness, but obviously this procedure cannot be used if ground temperatures are too shallow and active-layer thickness is unknown. Our procedure is applicable even in such situations.

The extrapolations have frequently been done rather intuitively and without a physical basis. We admit that the situation is somewhat better in the case of the active-layer thickness, but in principle none has been done for extrapolating the permafrost table temperature. The linear relationship between the thawing and freezing indices within the active layer is new, as is the whole procedure, which has a physical basis.

The ALT values were determined by continuous tracking/interpolating the 0 °C isotherm from measured temperatures (see e.g., Hrbáček et al., 2020, 2021; Kňažková and Hrbáček, 2024). The mean annual permafrost table temperature (MAPT) was the mean annual ground temperature interpolated to the depth corresponding to ALT (=permafrost table). We compared our model outputs directly with these measured/interpolated values.

Hrbáček, F., Cannone, N., Kňažková, M., Malfasi, F., Convey, P., Guglielmin, M. (2020). Effect of climate and moss vegetation on ground surface temperature and the active layer among different biogeographical regions in Antarctica. Catena, 190, 104562. https://doi.org/10.1016/j.catena.2020.104562

Hrbáček, F., Engel, Z., Kňažková, M., Smolíková, J. (2021). Effect of summer snow cover on the active layer thermal regime and thickness on CALM-S JGM site, James Ross Island, eastern Antarctic Peninsula. Catena, 207, 105608. https://doi.org/10.1016/j.catena.2021.105608

Kňažková, M., Hrbáček, F. (2024). Interannual variability of soil thermal conductivity and moisture on the Abernethy Flats (James Ross Island) during thawing seasons 2015–2023. Catena, 234, 107640. https://doi.org/10.1016/j.catena.2023.107640

**RC2:** The analysis and conclusions would benefit from better descriptions of the field sites including material properties, vegetation, climate etc. The limited range in their characteristics is a concern as all the sites are in cold permafrost of the continuous zone and likely tundra sites. This limits the conclusions that can be made regarding model performance and the broader applicability of the approach. Consideration of sites in warmer permafrost in the discontinuous zone including those in forested and peatland terrain would be useful as this would further back up statements regarding model performance including statements made regarding warmer permafrost.

**AC:** We more than doubled the number of the validation sites in the revised manuscript, which now cover much of the Earth's major permafrost regions (Antarctica, Arctic, Tibetan Plateau and high mountains) and offer us greater heterogeneity in terms of permafrost table temperature and active-layer thickness (approximately −19 °C to −0 °C

for MAPT and ~40 cm to ~300 cm for ALT), as well as in terms of surface cover, active-layer composition and stratigraphy, or permafrost zones. This makes the validations more robust. The summary table listing the key characteristics of the validation sites is provided in the appendix.

**RC2:** Additional comments related to the concerns raised above and other comments are provided below for the authors' consideration.

L21 – "indicators" might be better than "measures"

**AC:** Changed as suggested.

**RC2:** L22-23 – suggested revision: "Climate change has resulted in permafrost warming and active-layer thickening throughout the permafrost regions". Biskaborn et al. is now out of date with respect to the trends. I suggest you include Smith et al. (2024, State of Climate- Arctic), along with Noetzli et al. (2024), as it provides the details for Arctic and is up to date.

**AC:** Changed as suggested.

**RC2:** L29-30 – It is important to note that active layer thickness is not determined directly from geophysical surveys but is interpreted and it is difficult to determine ALT in warm permafrost with high unfrozen water contents.

**AC:** We totally agree and that is why we only state that geophysical surveys are among the methods used to investigate permafrost and active-layer thickness. We did not mention anywhere that active-layer thickness is determined directly using geophysics.

**RC2:** L27-34 – Smith and Brown (2009) outline the various methods used as does Streletskiy et al. (2022).

**AC:** Thank you for these useful references. We incorporateed them into the revised manuscript.

**RC2:** L34-37 – Even if ground temperatures are measured within shallow permafrost and the active layer the permafrost table temperature or active layer thickness still needs to be determined/calculated. Interpolation and extrapolation is the method usually used. I suggest you consult Riseborough (2008 ICOP) which describes the appropriateness of interpolation/extrapolation approaches.

**AC:** Thank you for this reference. We included it in the revised manuscript.

**RC2:** L38-42 – The purpose of the model is important. The ones mentioned are generally used for predictive applications such as determining conditions with little information on the site conditions. What you seem to be proposing is away to determining ALT or MAPT based on having ground temperature measurements which is what we do when we use interpolation/extrapolation of shallow ground temperatures to determine ALT or MAPT (see comment above).

**AC:** The models are primarily intended to be used for MAPT or ALT estimates at places where ground temperature measurements are too shallow and MAPT or ALT cannot be determined directly or where the spacing between temperature sensors is too large (there are many such stations). Among other possible applications, they can be used to establish typical values of thermal conductivity ratio (for MAPT) and edaphic factor (for ALT), which can then be used to model MAPT and ALT in the past or in the future. Note that the thermal conductivity ratio has never been estimated this way. Edaphic factor has been estimated using the near-surface thawing index and active-layer thickness, but obviously this procedure cannot be used if ground temperatures are too shallow and active-layer thickness is unknown. Our procedure is applicable even in such situations.

**RC2:** L45 – Is reference being made to air or ground temperatures here?

**AC:** It is meant to be both, because both air and ground temperatures are used to as model forcings (of course, for air temperatures, some procedures must be used to convert them to ground temperatures). We explicitly state this in the revised manuscript.

**RC2:** L46-49 – Some of the approaches mentioned do consider variable properties including thermal conductivity for thawed and frozen conditions (e.g. ratio between them is included in TTOP equation) or the variation of conductivity with temperature (and unfrozen water content).

**AC:** Yes, some (but few) of the approaches mentioned indeed considered the temporal variations in ground physical properties, but these variations would have to be involved in the models as additional forcings, which is frequently not available. Hence, most models treat the ground physical properties as constants, which brings complications in terms of their representativeness (for instance, one measurement per year or more should represent the temporal variations over this whole period). We believe that our solutions are useful in that they can simply address this general lack and/or non-representativeness of ground physical data.

**RC2:** L57 – Editorial suggestion – delete "Besides other solution (Garagulya, 1990)" – it is not adding anything.

**AC:** Remove from the revised manuscript.

**RC2:** L66 – Editorial suggestion – Delete first part of sentence: "Eq. (1) is also valid for temperatures measured…..layer, which is convenient because…."

**AC:** Changed as suggested.

**RC2:** L68 – Note that usually the reference to surface temperature measurements used in these equations is from a sensor in upper 3-5 cm of the ground so not using exact surface temperature. The surface temperature can be estimated using n-factor to provide input into the equation.

**AC:** We agree and consider this in the revised manuscript. However, the problems associated with ground surface temperature measurements (mentioned in the original manuscript) also relate to some extent to near-surface measurements.

**RC2:** L84 – If this is essentially extrapolation how is this different from the approach others use to determine the depth of the permafrost table and MAPT when they have temperatures at two or more depths?

**AC:** As detailed above, the major difference is that previous extrapolations have frequently been done rather intuitively and without a physical basis, which is not the case of our models.

**RC2:** L97 – Editorial suggestion - Delete first part of sentence: "ALT (m) can be calculated using the….

**AC:** Remove from the revised manuscript.

**RC2:** L99 – missing word: "…simplest form is as…."

**AC:** Changed as suggested.

**RC2:** L103 – See earlier comment regarding estimates of surface temperature used by others.

**AC:** We agree and consider this in the revised manuscript. However, the problems associated with ground surface temperature measurements (mentioned in the original manuscript) also relate to some extent to near-surface measurements.

**RC2:** L136 – Smith et al. (2009) is also relevant - used observed ALT from sites in various regions and environments (tundra, forest, peatland and mineral and organic soil) to show the range in the Edaphic factor.

**AC:** Thank you for this useful reference. We incorporated it in the revised manuscript.

**RC2:** L148 – See earlier comment regarding extrapolation

**AC:** The major difference is that previous extrapolations were frequently done rather intuitively and without a physical basis.

**RC2:** L159-197 – It seems that there are several assumptions being made as well as simplifications. For example, thermal conductivity changes with temperature due to change in unfrozen water but it appears that constant frozen and unfrozen conductivity are assumed. Are the results of the two models really comparable?

**AC:** Please note that we completely removed the numerical simulations for idealized scenarios from the revised manuscript, and the model evaluation now fully relies on the field observations, the dataset of which was more than doubled.

**RC2:** Table 2 – For years and seasons is the 2nd number the total record length and the first number the number of years utilized in analysis? It might be clearer to refer to number of years used and refer to it as e.g. 6 of 6. Are the ALT values determined from temperature or through probing? How is MAPT determined? It is unclear what you are comparing your modelled values with? I might have missed something here but maybe there needs to be a clearer explanation

**AC:** W admit this was not very intuitive and we reworked it to make it fully understandable in the revised manuscript.

The ALT values were determined by continuous tracking/interpolating the 0 °C isotherm from measured temperatures (see e.g., Hrbáček et al., 2020, 2021; Kňažková and Hrbáček, 2024). The mean annual permafrost table temperature (MAPT) was the mean annual ground temperature interpolated to the depth corresponding to ALT (=permafrost table). We compared our model outputs directly with these measured/interpolated values.

**RC2:** L200-201 – No information on the sites is provided so the reader doesn't know how diverse they are. There is no information provided on material characteristics or vegetation. The Alaskan sites are all on the North Slope in the continuous permafrost zone and likely in tundra environments, so conditions are not that diverse with respect to climate and vegetation. Using field data from sites in warmer permafrost in discontinuous zone and for forested sites would provide more diverse conditions. This would help show if your approach is valid for a wide range in conditions.

**AC:** We more than doubled the number of the validation sites in the revised manuscript, which now cover much of the Earth's major permafrost regions (Antarctica, Arctic, Tibetan Plateau and high mountains) and offer us greater heterogeneity in terms of permafrost table temperature and active-layer thickness (approximately −19 °C to −0 °C for MAPT and ~40 cm to ~300 cm for ALT), as well as in terms of surface cover, active-layer composition and stratigraphy, or permafrost zones. This makes the validations more robust. The summary table listing the key characteristics of the validation sites is provided in the appendix.

**RC2:** L204 – Since you are referring to a depth it would be better to refer to permafrost table or base of the active layer. Do you mean the base of the active layer was above the shallowest sensor?

**AC:** Please note that we used temperatures measured at the depth intervals of 0−10 cm, 25−35 cm and 45−55 cm as model forcings so that they are comparable across the study sites, but the depths at each site were constant, for instance, 5 cm, 30 cm and 50 cm. If the active layer is thin, there can be some years when the base of the active

layer was shallower than the deepest sensor used. This means that the active-layer thickness was less than 50 cm. But definitely, we did not mention anywhere that the base of the active layer was above the shallowest temperature sensor.

**RC2:** L206-2011 – See Riseborough et al. (2008 ICOP) regarding errors associated with different approaches (interpolation, extrapolation) to determine thaw depth/top of permafrost etc. and guidance on the best approach to use.

**AC:** Thank you for this reference. However, please note that we used linear interpolation based on measured temperatures to determined MAPT and ALT, which we also used in numerous previous publications.

**RC2:** L222 – Wasn't this exponential decrease already fairly well known? Doesn't the magnitude of the decrease depend on the material properties?

**AC:** It was known for thawing indices (e.g., Riseborough, 2003), but rather neglected for freezing indices because permafrost studies have dominantly focused on summer active-layer dynamics and/or annual means. However, the linear relationship between the thawing and freezing indices within the active layer is new. And yes, the magnitude of the decrease mostly depends on the thermal conductivities and the amount of latent heat.

Riseborough, D. (2003). Thawing and freezing indices in the active layer, in: Proceedings of the 8th International Conference on Permafrost, Zurich, Switzerland, 21–25 July 2003, 953-–958, 2003.

**RC2:** L235 – See earlier comment – How were observed values determined?

**AC:** The observed MAPT values were determined by a linear interpolation of the mean annual ground temperatures observed at sensors just above and below the observed active-layer thickness, which was briefly described in Sect. 3.3. We made this a bit clearer in the revised manuscript so that it is easily understandable (now it is in Sect. 3).

**RC2:** L260 – See earlier comment – How were observed values determined?

**AC:** The ALT values were determined by continuous tracking/interpolating the 0 °C isotherm from measured temperatures (see e.g., Hrbáček et al., 2020, 2021; Kňažková and Hrbáček, 2024), which was briefly described in Sect. 3.3. We made this a bit clearer in the revised manuscript so that it is easily understandable (now it is in Sect. 3).

**RC2:** L265-268 – Does the difference in error between Antarctica and Alaska sites have anything to do with the material properties. Was latent heat more of a factor for the AK sites?

**AC:** We believe that it does. We think this was mainly caused by the active-layer stratigraphy, which is almost exclusively one-layer without any organic material at bare-ground sites, while it is frequently two-layer at vegetated sites.

**RC2:** L264-292 – The thermal offset depends on the ratio of thawed and frozen thermal conductivity which depends on the amount of moisture/ice in the ground. If the moisture content is low or arid conditions exist, then the offset will be low or positive. Is the site in McMurdo Sound a dry site? It would be useful to know this. It would have been good to use sites with warmer permafrost in your analysis to back up the comment that deviation in MAPT estimates would be larger.

**AC:** Yes, the sites in McMurdo Sound experience hyperarid conditions, which most likely produce positive thermal offsets. Please note that we more than doubled the number of the validation sites in the revised manuscript, which now cover much of the Earth's major permafrost regions (Antarctica, Arctic, Tibetan Plateau and high mountains) and offer us greater heterogeneity in terms of permafrost table temperature and active-layer thickness (approximately −19 °C to −0 °C for MAPT and ~40 cm to ~300 cm for ALT), as well as in terms of surface cover,

active-layer composition and stratigraphy, or permafrost zones. Generally, they back up our statement that larger deviations in MAPT tend to occur in warmer conditions.

**RC2:** L306-310 – It would be useful to have information on the material properties at the field sites to back up these statements.

**AC:** We totally agree that it would be useful, but unfortunately the information on the material properties are scattered or rather general/descriptive from the validation sites. Hence, we can draw only general conclusions in this respect. If we wanted to have detailed information on the material properties, the number of validation sites would have to be much smaller, which is, however, undesirable in terms of robustness of the validations.

**RC2:** L315-330 – Although these other approaches make assumptions regarding thermal properties etc. based on general site characteristics, information on ground temperature is not required and the models determine the ground temperatures. This makes them useful for determining current and future conditions. This might make them more broadly applicable.

**AC:** Unfortunately, we do not understand this comment clearly. Anyway, it is important to note that all other models for permafrost table temperature and active-layer thickness require information on ground (surface or near-surface) temperature, which is used as their upper boundary condition.

**RC2:** L327-330 – If temperature below the permafrost table was available would it be used if there were only one sensor at a shallower depth? You state that inputs can be any depth combination within the active layer based on temperature data availability and site characteristics. What are the site characteristics being referred to?

**AC:** Unfortunately, temperatures below the permafrost table cannot be used because thawing indices must have non-zero (=positive) values; otherwise the outputs would be erroneous.

By the site characteristics we meant that specific depth combinations may work better under specific site characteristics.

**RC2:** L331-334 – Aren't these products based on modelling with various assumptions made regarding ground properties etc.

**AC:** Yes, but these assumptions and/or ground properties are largely unknown for these products, which considerably impedes model applications. Consequently, for instance, the active-layer thickness estimates are limited by the deepest ground temperature level (node) available in these products, which is frequently shallow and situated within the active layer. However, our models can deal with such situations. We emphasized this a bit further in the revised manuscript.

**RC2:** L338-340 – This is likely one of the primary sources of error especially with respect to moisture/ice contents and latent heat effects as discussed in Riseborough (2003).

**AC:** Yes, it certainly is, as in any analytical model, and we think it is only fair to admit it.

**RC2:** L340-348 – Riseborough (2008) is probably relevant here especially with respect to spacing of temperature measurements etc. in determining thaw depth.

**AC:** We think this is much more relevant, for instance, to interpolating the active-layer thickness or calculating the permafrost table temperature from measured ground temperatures. In terms of this paper, this is therefore particularly relevant to the validation data used to evaluate the models. However, the models themselves are in principle independent of the spacing of temperature measurements. More important is the sensor position with respect to the active-layer stratigraphy.

**RC2:** References

Riseborough, D.W. 2008. Estimating active layer and talik thickness from temperature data: implications from modeling results. In Ninth International Conference on Permafrost. Edited by D.L. Kane and K.M. Hinkel. Fairbanks, Alaska. Institute of Northern Engineering, University of Alaska Fairbanks, Vol.2, pp. 1487-1492.

Smith, S.L., Romanovsky, V.E., Isaksen, K., Nyland, K., Shiklomanov, N.I., Streletskiy, D.A., and Christiansen, H.H. 2024. Permafrost (Arctic) [in "State of the Climate in 2023"]. Bulletin of the American Meteorological Society (supplement), 105(8): S314-S317. doi:10.1175/BAMS-D-24-0101.1

Smith, S.L., Wolfe, S.A., Riseborough, D.W., and Nixon, F.M. 2009. Active-layer characteristics and summer climatic indices, Mackenzie Valley, Northwest Territories, Canada. Permafrost and Periglacial Processes, 20(2): 201-220. doi:10.1002/ppp.651

Smith, S. and Brown, J. 2009: Assessment of the status of the development of the standards for the Terrestrial Essential Climate Variables - T7 - Permafrost and seasonally frozen ground. GTOS 62 Essential Climate Variables

Streletskiy et al. 2022 Measurement Recommendations and Guidelines for the Global Terrestrial Network for Permafrost (GTN-P). DOI: 10.5281/zenodo.5973079

---

## Author Response (AR2)

**AUTHORS' RESPONSE TO THE COMMENTS OF THE EDITOR**

Dear Tomáš Uxa and co-authors,

as you have certainly noticed two reports from the second review round are online. While one of them mainly requests to include permafrost sites in forest areas, the other report is very critical. It raises four major comments - all valid and constructive in my opinion. First of all, your MAPT and ALT 'observations' seem to be generated by linear interpolation. This appears to heavily undermine the validation of your model. Second, reviewer #3 points you to a valuable database for circum-arctic active layer monitoring. Third, the reviewer requests you to assess the performance of a baseline model to better understand the benefit of your approach. Last, you are pointed to two other approaches (Nicholas, 1995; Romanovsky, 1996) that also indirectly infer bulk material properties of permafrost similar to your method. This calls for a moderation when presenting your method.

In summary, I consider the comments on the validation and the baseline model as crucial for continuing to consider your article for publication in TC. Despite the reviewer recommendation, I invite you to submit a revised manuscript. If you should see yourself incapable of addressing these comments, please consider to re-submit your manuscript as a shorter or more technical report.

Best,
The editor, Johannes Fürst

Dear Editor,

On behalf of co-authors, I am submitting the second revision of the manuscript ID EGUSPHERE-2024-2989 entitled "Simple analytical–statistical models (ASMs) for mean annual permafrost table temperature and active-layer thickness estimates" by Tomáš Uxa, Filip Hrbáček, and Michaela Kňažková.

We responded point-by-point to all the reviewers' comments and suggestions and made corresponding revisions in the manuscript. Our responses are included at the end of this text, starting on page 2. Please note that the reviewers' comments are prefixed by bold RC (1 or 3), while the authors' comments are prefixed by bold AC. All the reviewers' comments are also in blue in order to distinguish them from the authors' responses.

We included additional 11 forest and 1 shrub site so that their proportion of the total dataset (now counting 55 sites) has substantially increased in the revised manuscript. Correspondingly, we also recalculated the results and revised the text, tables and figures. However, the results changed to minor extents, which suggests that the analysis was already robust enough.

The referee #3 mostly criticizes our approach, but we lack any kind of constructive suggestions and specific recommendations on how we should improve the manuscript. Therefore, in our reply, we respond to the particular points of the criticism and unsubstantiated statements that mostly concern the validation of the proposed models. However, we performed the validation using standard approaches for MAPT/ALT determination, which are widely accepted by the research community. If this approach was considered illegitimate or seriously flawed, it would call into question the majority of results published to date on the active-layer thermal regime. The major points of our response to the referee #3 (mostly related to the MAPT and ALT for validation) were also incorporated into the revised manuscript, especially Sect. 3 "Model evaluation" (please see our detailed response).

We believe that the improvements made in the revised manuscript are sufficient.

Thank you very much for considering the revised manuscript.

Yours sincerely,
Tomáš Uxa

**AUTHORS' RESPONSE TO THE COMMENTS OF THE REFEREE #1**

**RC1:** The manuscript has been significantly improved by the revision work. The presentation and discussion of the proposed methodology is clearer, and the validation is more convincing. Meanwhile I have a few minor concerns regarding the discussion of the underlying assumptions and the chosen set of validation sites, see my comments below. Thus I recommend a minor revision of the manuscript prior to its publication in TC.

**AC:** Thank you for your review.

**RC1:** l 255-256: "Although ASMs utilize only thawing and freezing indices from two depth levels within the active layer as inputs, they inherently account for the natural variability of ground physical properties in the intermediate layer between these two depths." I understand better now what the authors mean here, nevertheless I think that it should be highlighted that with the developed methodology the variability of ground physical properties is handled by means of an averaging operation, resulting in a temporally (seasonal) and spatially (within all the layer between z1 and z2) average of kt/kf.

**AC:** We revised the sentence as follows: "Although ASMs utilize only thawing and freezing indices from two depth levels within the active layer as inputs, they inherently account for the natural variability of ground physical properties in the intermediate layer between these two depths that is expressed in terms of annual and seasonal means of the thermal conductivity ratio and edaphic term, respectively."

**RC1:** l 269-270: "Of course, ASMs in principle also treat them as constants, but their values are representative for individual years (Eq. 8) or thawing seasons (Eq. 27)" Here the sources of temporal variability (e.g.: temporal variation of water content) should be explicated.

**AC:** We revised the sentence as follows: "Of course, ASMs also treat them as constants, but their values are annual or seasonal means that reflect the variations in ground physical properties over time mainly due to changes in water content and as such they are representative for individual years (Eq. 8) or thawing seasons (Eq. 27). This is a major improvement over other analytical or statistical models…"

**RC1:** Table C1 – the increase in number and diversity of the considered validation sites is a strong improvement of the manuscript. Nevertheless the proportion of forested sites is small (5/43, ~12%) compared to the proportion of the permafrost area covered by boreal forest (55%, Stuenzi et al., 2021). Additional sites in boreal forest areas should be added, or at least the possible biases linked to the used sampling of permafrost areas conditions should be discussed.
Stuenzi, S. M., Boike, J., Gädeke, A., Herzschuh, U., Kruse, S., Pestryakova, L. A., Westermann, S., and Langer, M.: Sensitivity of ecosystem-protected permafrost under changing boreal forest structures, Environ. Res. Lett., 16, 084045, https://doi.org/10.1088/1748-9326/ac153d, 2021.

**AC:** We included additional 11 forest and 1 shrub site so that their proportion of the total dataset (now counting 55 sites) has substantially increased in the revised manuscript. However, note that the results for shrub and forest sites as well as the total ones changed to minor extents, which suggests that the analysis was already robust enough.

**AUTHORS' RESPONSE TO THE COMMENTS OF THE REFEREE #3**

**RC3:** In this paper, the authors propose set of semi-analytical models for estimating active layer thickness (ALT) and mean annual permafrost temperatures (MAPT) in permafrost-affected soils using only ground temperatures measured in the active layer. The presented equations are relatively straightforward algebraic derivations from well known equilibrium models of permafrost temperature (TTOP) and thaw (Stefan model). The authors validate their approach using ground temperature data from 43 sites in the Arctic spanning a wide range of surface conditions.

**AC:** Thank you for your review, even though it mostly criticizes our approach. Unfortunately, we lack any kind of constructive suggestions and specific recommendations on how we should improve the manuscript. Therefore, in our reply, we aim to provide you with a thorough response to the particular points of your criticism and unsubstantiated statements. The major points of our response (mostly related to the MAPT and ALT for validation) were also incorporated into the revised manuscript, especially Sect. 3 "Model evaluation".

Please note that our validation dataset covers a much wider area than the Arctic and a broader range of permafrost environments. In addition to the Arctic, it includes other Earth's major permafrost regions such as Antarctica, Qinghai-Tibetan Plateau and European Alps (now counting 55 sites), where high-quality ground temperature measurements are available. Although this number may seem low, it reflects the fact that the sites were carefully selected with respect to data availability, overall continuity of datasets, and sensor spacing.

**RC3:** It is unclear to me whether or not the equations presented by the authors in this work are truly novel. I have not seen them before in this particular form anywhere in the literature, so I will give them the benefit of the doubt and assume that they are. However, the derivations are relatively straightforward and follow from long established equilibrium models, so I think that the added value is relatively low, all other assumptions and limitations of such equilibrium models notwithstanding.

**AC:** It is true that the analytical-statistical models for MAPT and ALT given by Eq. (8) and (27) were derived from long-established models (TTOP and Stefan models). However, to our knowledge, Eq. (8) and (27) themselves have never been published so far anywhere (papers, monographs, or conference proceedings) and thus they are totally novel. We also believe that if any derivation is simple and straightforward, this is an advantage rather than the opposite. No matter what it is based on, if it is new.

The added value of our solutions is a matter of largely subjective opinion, but we respectfully disagree with the statement that "all other assumptions and limitations" of the novel models are the same as those of the TTOP and Stefan models. Our solutions address the issue of the utilization of the ground physical properties for the modelling. They reflect the variations in ground physical properties over time mainly due to changes in water content and as such they are representative for individual years (Eq. 8) or thawing seasons (Eq. 27). This is a major improvement over other analytical or statistical models.

**RC3:** The equations themselves are indeed perhaps useful in some problem settings and would certainly fit well in a technical report. However, I do not think that the level of novelty or scientific rigor of the validation meets the editorial standard for a full journal article.

From the scientific perspective, this article has two major flaws:

1. The "observed" MAPT and ALT are not actually observations, but are rather obtained by linear interpolation of observed ground temperatures in both the active layer and permafrost via linear interpolation. As far as I could tell, the authors do not report the depths of the temperature sensors which were used to calculate these pseudo-observations, which is also problematic. Such approximations could potentially be acceptable if the interpolation is over a short distance (<10 cm or so) but otherwise cannot be expected to realistically determine the actual MAPT and ALT.

**AC:** Thank you for your opinion, however, we disagree that we are generating "pseudo-observations". The interpolation of ALT is widely used and accepted and in important permafrost environments such as mountains or

high polar regions (extensive occurrence of exposed bedrock surfaces or gravelly to blocky materials) it is the only method for ALT estimates because manual probing is impossible. Further, if we are seeking the position/depth of 0 °C isotherm in the particular borehole/profile (site) there is no other way than using and interpolating data from ground temperature measurements as these cannot be spatially continuous. By contrast, there is always some sensor spacing that tends to increase with depth and interpolating between them is therefore a standard practice (see e.g. Measurements Recommendations and Guidelines at Global Terrestrial Network for Permafrost portal at http://gtnpdatabase.org).

To our knowledge, there is no standard so far, which recommends particular interpolation method or spacing between thermometers including the justification why this particular method is better than others. However, Measurements Recommendations and Guidelines at Global Terrestrial Network for Permafrost (at http://gtnpdatabase.org) suggest linear interpolation and sensor spacing within the active layer of 0.2, 0.4, 0.8, 1.2, 1.6, 2, 2.5 m). The contribution by Riseborough (2008) deals with the possible inaccuracies of the interpolation, but unfortunately these ideas were not developed further, remaining this issue unresolved. It is also necessary to state that the assumption of Riseborough (2008) is solely based on modelling data and have not been verified in any way in the field. Yet, considering this general knowledge, the vast majority of localities was selected so that the maximum distance between sensors was lower than 25 cm and 50 cm (for ALT <100 and >100 cm, respectively; please note that this is much finer depth resolution/smaller sensor spacing than recommended – see above), which on the one hand significantly reduced the number of datasets available for the validation (sensor spacing is greater at numerous sites), but on the other hand increased the accuracy of the observed MAPT and ALT. However, several exceptions were made to this rule in order to balance the different surface covers in the validation dataset.

Regarding the use of the interpolated temperature to estimate MAPT, the error is much lower in comparison to ALT. Usually, there is a low gradient in the mean annual ground temperature near the base of the active layer/permafrost table and so the temperature difference observed at the deepest sensor in the active layer and at the topmost sensor in permafrost was lower than 0.2 °C. In other words, if we would use measured data from the first (=uppermost) sensor located in permafrost only the overall effect on the accuracy of the MAPT estimate would be negligible.

We quantified and included in the revised manuscript that the temperature differences between the observed MAPT and the temperature of the closest sensor used for the interpolation differed by less than 0.1 °C in ~65 % of cases and by less than 0.2 °C in ~90 %. The distance between the observed ALT and the closest temperature sensor used of the interpolation was less than 10 cm in ~80 % of cases and less than 20 cm in almost 100 %. This gives the maximum possible deviations of observed MAPT and ALT from their actual values. We therefore think that the accuracy should be mostly acceptable also based on the reviewer's opinion (=when interpolation is over a short distance [<10 cm or so]).

**RC3:** This also likely explains why the model appears to perform so well at many of the sites. According to the authors' own description, their method effectively amounts to a form of linear extrapolation from the two measured active layer temperatures. By generating pseudo-observations via linear interpolation, the authors are effectively generating data which, by construction, will be well predicted by such a linear extrapolation, in most cases. It is unclear why the authors did not attempt to use actual active layer measurements such as those from the Circum-arctic Active Layer Monitoring (CALM) network.

**AC:** The utilization of data from CALM sites is not feasible for this kind of validation from following reasons:

1) Please note that the MAPT and ALT estimates using Eq. (8) and (27) were exclusively based on the depth pairs from within the active layer and especially its uppermost part in most instances. By contrast, the observed MAPT and ALT was determined using the linear interpolation between the deepest sensor in the active layer and at the topmost sensor in permafrost. This means that the sensors used to determine the observed and modelled values differed in all instances! There was not a single identical pair of sensors used for both validation and modelling so your assumption is incorrect.

2) The data presented in the CALM database are typically mean values from (usually) 121 measurements within grids of 100 x 100 m, which typically show a high spatial variability within the grid, as was documented in studies from different Arctic as well as Antarctic regions. Even if we would have ground temperature data from

boreholes at the individual CALM sites, we could not validate active-layer thickness determined from borehole data against mean thaw depth from the grid of 100 x 100 m.

3) Manual thaw-depth probing measures a physical state of the ground using rigid rods that are pushed vertically into the ground to the depth at which ice-bonded material provides firm resistance. Since ice formation at sub-zero temperatures is complex and ground freezing characteristic curves differ between substrates, manually determined thaw depth may not necessarily correspond to the position of 0 °C isotherm, which is obtained by models.

4) Manual thaw-depth probing is impossible in mountains or high polar regions with extensive occurrence of exposed bedrock surfaces or gravelly to blocky materials. Consequently, there are no manual thaw-depth measurements from such sites.

5) Numerous CALM sites, such as those from Svalbard, Switzerland or Mongolia, therefore only utilize ground temperature measurements at depths of 0, 1, 2, 3 m and their subsequent interpolation to determine ALT (see https://www2.gwu.edu/~calm/data/north.htm). Hence, a substantial part of the CALM database, which you recommend to use for model validations, also generates "pseudo-observations" and should be considered unreliable according to your opinion.

6) We think that using and interpolating ground temperature measurements makes the validation dataset homogeneous because uniform methods can be used across all sites. This would be impossible for CALM sites where thaw-depth is measured manually by different persons and frequently only once a year, which means that the maximum thaw depth (=active-layer thickness) is in reality rarely reported from these sites. This is unsuitable for validations of our models.

**RC3:** 2. Although the validation is ostensibly extensive in terms of the number of field sites, the authors provide no baselines for comparisons. This is especially surprising considering that the authors' method is derived from the TTOP and Stefan models, both of which would serve as natural baselines. The authors might respond that the novelty of their method is in the fact that they do not require estimates of the frozen vs. thawed thermal conductivities, information which they do not have for many or all of the sites.

**AC:** Indeed, this is the major reason why we cannot use the TTOP and Stefan models themselves for the validations. To our knowledge, the data on thermal properties, but also on ground water content, are simply unavailable for the validation sites, which makes direct validation against the two models impossible. Further, using TTOP/Stefan models as reference would provide us with the accuracy of ASMs against these two models but not against observed data.

**RC3:** However, this would seem to ignore the fact that methods such as that of Hinkel and Nicholas (1995) and Romanovsky (1996), both of which are cited within, get around this simply by treating lumping these unknown terms into a linear parameter which can be estimated directly from the data. It would also be natural for the authors to compare their method to the similar and widely used approach of extrapolating the MAGT at the two measured points down to the zero-degree isotherm, which they themselves state is closely related.

**AC:** Please note that the edaphic term does not fully solve the issue of missing parameter. Since it combines the information on thermal and moisture properties, it also shows natural year-to-year variations caused by the variability of these ground properties. This means that it in principle leads to random over/underestimation of ALT over time, while our procedure considers the actual (average) state of the active layer in individual years or thawing seasons. Notably, the edaphic term is impossible to use at sites where you lack any data on ALT, which are crucial for the calculation of the edaphic term. This is one of the issues which we suggest to solve using ASMs.

**RC3:** In summary, although the equations derived by the authors may be of some interest to polar researchers and practitioners in the form of a technical report, the scientific novelty is modest at best, and the rigor of the validation is seriously lacking. Given that the paper has already gone through a full round of revision, I would recommend rejection.

**AC:** Your main criticism concerns the validation of the proposed ASMs. However, we performed the validation using standard approaches for MAPT/ALT determination, which are widely accepted by the research community. If this approach was considered illegitimate or seriously flawed, it would call into question the majority of results published to date on the active-layer thermal regime. We believe that we addressed and explained these issues sufficiently.

---

## Author Response (AR3)

**AUTHORS' RESPONSE TO THE COMMENTS OF THE EDITOR**

Dear Tomáš Uxa and co-authors,

I have carefully read through your response to the reviewer comments. You have added 11 forest and 1 shrub sites to your final list of 55 evaluation sites of permafrost monitoring. Concerning the critical comments of reviewer #3, I want to summarise your answers to each of the 4 major comments. First and with regard to the 'pseudo-observations', you now specify the sensor spacing as well as statistics on temperature differences and vertical distance between your interpolated observations and the closest sensor. Second, you explain that the CALM measurements are not suitable for a comparison due to the gridding, the measurement protocol and the often large vertical spacing of temperature sensors. Third, a baseline model comparison would require assumptions on unknown material properties (advantage of your method), which seems a somewhat vain exercise. Fourth, I am less convinced by your argument to reject a comparison to other approaches indirectly inferring bulk material properties. You state that the edaphic term for these methods can only be inferred if the ALT is measured. Please specify why ALT measurements are not available at your monitoring sites (the CALM database would hold such estimates, I guess). I request a firm argument why such a model comparison was not possible. If feasible, consider to present a brief comparison even if it was only for a few selected sites.

In summary, I continue to consider your manuscript for publication in TC and invite you to answer to my comment above (with a revised manuscript). This will only require a minor revision round.

Best,
The editor, Johannes Fürst

Dear Editor,

On behalf of co-authors, I am submitting the third revision of the manuscript ID EGUSPHERE-2024-2989 entitled "Simple analytical–statistical models (ASMs) for mean annual permafrost table temperature and active-layer thickness estimates" by Tomáš Uxa, Filip Hrbáček, and Michaela Kňažková.

Thank you for reading our previous response to the reviewer reports and commenting on it.

With regard to the fourth point of the reviewer #3. In fact, we already compared the thermal conductivity ratios and the edaphic terms modelled using Eq. (5) and (20) for the three depths pairs of 5/30 cm, 5/50 cm and 30/50 cm against these bulk material properties for the whole active layer inferred by rearranged Eq. (2) and (23) based on the observed MAPT, ALT and thawing and freezing indices for the uppermost available temperature sensors. Instead of the CALM database, however, we did this based on our validation dataset consisting of 55 locations. This is illustrated in Figures 4 and 5, and since it is a kind of by-product of the MAPT and ALT estimates that constitute the central part of our manuscript, it is only mentioned and discussed in several paragraphs of the Discussion on lines 231–240, 258–265 and 270–298. We also revised the text there slightly and provided some new citations. In addition, we included error statistics in Figures 4 and 5.

We believe that these clarifications and minor revisions of the manuscript are sufficient.

Thank you very much for considering the revised manuscript for publication in TC.

Yours sincerely,
Tomáš Uxa